



# Integrated System for Atmospheric Boundary Layer Height Estimation (ISABLE) using a Ceilometer and Microwave Radiometer

Jae-Sik Min, Moon-Soo Park, Jung-Hoon Chae, and Minsoo Kang

Research Center for Atmospheric Environment, Hankuk University of Foreign Studies

*Correspondence to*: Moon-Soo Park (ngeograph2@gmail.com)

**Abstract.** Accurate boundary-layer structure and height are critical in analyzing the features of air pollutants as well as the local circulation. Although surface-based remote sensing instruments give high temporal resolution of the boundary-layer structure, there were many uncertainties in determining the accurate atmospheric boundary-layer heights (ABLHs). In this study, an algorithm for an integrated system for ABLH estimation (ISABLE) was developed and applied to the vertical profile
data obtained by a ceilometer and microwave radiometer in Seoul City, Korea. A maximum of 19 ABLHs were calculated by the conventional time-variance, gradient, wavelet, and clustering methods using the backscatter coefficient data. Meanwhile, several stable boundary layer heights were extracted by the near-surface inversion and environmental lapse rate methods using potential temperature data. The ISABLE algorithm could find an optimal ABLH from post-processing including k-means clustering and density-based spatial clustering of applications with noise (DBSCAN) techniques. It is found that the ABLH
determined by ISABLE exhibited larger correlation coefficients and smaller absolute bias and root mean square errors between the radiosonde-derived ABLHs than those obtained by most conventional methods. Clear skies showed higher daytime ABLH than cloudy skies, and the daily maximum ABLH was recorded in spring due to the stronger radiation. The ABLHs estimated by ISABLE are expected to contribute to the parameterization of vertical diffusion in the atmospheric boundary layer and in understanding the severe smog/haze events that arise from by fumigation during the ABL evolution period.

**1 Introduction**

Atmospheric boundary layer (ABL) is the lowest part of the troposphere that is directly influenced by the earth's surface (Garratt, 1994). The ABL is repeated in a daily cycle with a mixed-layer (ML) that mixes vertically by the convection induced by surface heating or wind shear in the daytime and a stable boundary-layer (SBL) or residual layer that appears in the upper atmosphere as the lower atmosphere is stabilized by the cooling of the surface at night. The ML is an important meteorological
factor that affects the vertical mixing and transport of air pollutants by convection and turbulence. Especially, it can be used as an important factor in estimating the diffusion of pollutants near the surface (Stull, 1988). Besides, when SBL exists at night, the lower atmosphere is stabilized and stagnant, and atmospheric diffusion does not occur in the lower layer, resulting in higher concentration (Stull, 1988; Emeis and Schäfer, 2006). Thus, the ABL can be used as a meteorological factor to determine the



air pollutant concentration, and the determination of ABL height (ABLH) is very important for understanding the transport

process of the lower atmosphere (Garratt, 1993).

Many previous studies have developed various methodologies for determining ABLH, such as the ML height (MLH) and SBL height (SBLH). The ABLH has traditionally been determined using in-situ radiosonde data. The parcel method using the vertical profile of virtual potential temperature (Holzworth, 1964; Seibert et al., 2000) and the gradient method using the vertical gradient of the virtual potential temperature or mixing ratio are widely used (Oke, 1987; Stull, 1988). Alternatively,

ABLH can be determined by the Bulk Richardson number, which includes the thermal buoyancy term generated by surface heating and the turbulent term arising from the vertical wind shear (Vogelezang and Holtslag, 1996; Zilitinkevich and Baklanov, 2002; Zhang et al., 2014). Radiosonde produces vertical in-situ data and was used as a reference in many previous studies (e.g., Eresmaa et al., 2006; Basha and Ratnam, 2009; Collaud Coen et al., 2014). However, there is a limit that clearly distinguishes ABLH from radiosonde observations (Seibert et al., 2000). When the solar radiation energy is strong in the daytime, the

distinction of the boundary layer such as the convective boundary-layer is obvious, but at other times the layer separation is not clear. In addition, the major drawback of radiosonde is that it has a coarse temporal resolution (6−12 hours) to determine ABLH with the diurnal variation (Schween et al., 2014).

In the past two decades, several researchers have determined ABLH using surface-based remote sensing instruments to overcome the drawbacks of temporal resolution. The aerosol lidar and ceilometer produce a vertical profile of the

backscattering coefficient, which is scattered from a ground-launched laser back into the atmospheric aerosol. The measured backscattering coefficient can be used to analyze the features of vertical aerosol distribution, and the ABLH can be determined through the separation of aerosol layers. In the ML, vertical mixing of the aerosol particle is active, and the backscattering coefficient is relatively homogeneous, whereas it decreases sharply at the above the MLH. Using this property, the gradient method, which designates the altitude with the maximum vertical gradient of the backscattering coefficient as ABLH (e.g.

Flamant et al., 1997; Sicard et al, 2005; Lammert and Bösenberg, 2006; Münkel et al., 2007; Emeis et al., 2008; Summa et al., 2013; Schween et al., 2014), and the wavelet method, which determines ABLH as the altitude at which the wavelet covariance coefficient is maximum (e.g. Gamage and Hageberg, 1993; Cohn and Angevine, 2000; Brooks, 2003; Morille et al, 2007), are most widely used. Menut et al. (1999) analyzed the ABL structure by using the inflection point method (second derivative method) and centroid method (time-variance method) to understand the chemical and physical processes involved in pollution

events in Paris. The growth and attenuation of ABLH is repetitive due to the heating and cooling of the surface with the temporal variation in solar and terrestrial radiations. As a result, the vertical aerosol distribution in the aerosol layer changes with time, and the ABLH can thus be determined by the time-variance of aerosol temporal distribution. Toledo et al. (2014) determined ABLH as a classification of the distribution of the backscattering coefficient value whose vertical profile is rapidly decreased or increased using k-means clustering. Furthermore, ABLH was estimated using an extended Kalman filter (EKF)

(Lange et al., 2014; Lange et al., 2015; Saeed et al., 2016). The EKF technique could be used in low signal-to-noise ratio (SNR) atmospheric scenarios without the need for long-time averaging and range smoothing (Caicedo et al., 2017; Dang et al., 2019).



Even though several methods have been developed, no consensus on a specific algorithm has been reached yet (Schween et al., 2014). Different methodologies provide different results of the ABLH depending on the weather condition and phenomena. Therefore, in this study, ABLH was determined using the methodologies developed in the previous studies. For this purpose,

we developed an integrated system for ABLH estimation (ISABLE) using statistical techniques to produce one ABLH optimized by combining the determined ABLHs by previous methodologies.

This paper is organized as follows. Section 2 introduces the observation station and instruments used in this study, section 3 describes the used data and preprocessing, and section 4 describes the ABLH estimation methods and ISABLE characteristics. In section 5, the ABLH estimation results are compared with the radiosonde-derived ABLH, and the diurnal variation features

are described. Finally, the summary and discussion are presented in section 6.

## 2 Site and Instrumentation

In this study, we used a ceilometer and a microwave radiometer, and a net radiometer installed at Jungnang station (127.08°E, 37.59°N, 45 m; Fig. 1), a super site of UMS-Seoul (urban meteorological observation system network in the Seoul metropolitan area; Park et al., 2017). The Jungnang station is located in Seoul City, Korea, and the surrounding buildings form an

environment that can be classified as a dense urban residential area with homogeneous heights (Park, 2018). This location is classified as UCZ−2 (intensely developed high density) according to the UCZ (urban climate zone) classification of Oke et al. (2004), and LCZ−2E (compact mid-rise, bare rock or paved) according to the LCZ (local climate zone) classification of Stewart and Oke (2012).

The ceilometer (model: CL51, manufacturer: Vaisala) produces a real-time vertical profile of backscattering coefficients each

minute at 10 m intervals up to 15,400 m a.g.l. using a laser (InGaAs diode laser) of 910 nm wavelength. It also measures the cloud base heights of three layers up to 13,000 m and the five-minute mean cloud cover in a one-minute interval.

The microwave radiometer (model: HATPRO−G4, manufacturer: RPG) observes atmospheric attenuation and brightness temperature from electromagnetic radiation emitted from the atmosphere using 14 channels (22−31 GHz, 7 water vapor channels; 51−58 GHz, 7 temperature channels). The measured atmospheric attenuation and brightness temperature are

converted to a vertical profile of atmospheric temperature, relative humidity, and liquid water path using a neural network model. The microwave radiometer produces two types of temperature profiles, namely, zenith measurements for the entire troposphere (0−10 km), and elevation scanning that provides an enhanced vertical resolution within the boundary layer (0−2 km). The temperature profiles of the two types are obtained by merging to a single profile. The vertical resolution is denser in the lower layer, but it decreases with height (30 m up to 1.2 km, 200 m up to 5 km, and 400 m up to 10 km), and a profile is

produced every ten minutes.

The ABLH was estimated using the vertical profiles of the backscattering coefficient from the ceilometer and the potential temperature from the microwave radiometer. The radiation observations obtained by the net radiometer were used to classify ABLH as daytime and nighttime values.





## 3 Data and pre-processing

**3.1 Radiosonde experiment**

Radiosonde is widely used to verify surface-based remote sensing instruments because it directly observes the meteorological parameter with height, and is used in this study to verify the results of ABLH estimation. Radiosonde measures vertical profiles of temperature, relative humidity (or mixing ratio), wind direction and speed, and pressure. The vertical profile of potential temperature and virtual potential temperature can be calculated using the observed meteorological variables.

Seoul City is affected by local circulation such as sea-land and mountain-valley breezes due to the presence of Yellow Sea and mountainous terrain (Park and Chae, 2018). In order to analyze the structure of the atmospheric boundary layer in urban areas, 171 radiosonde sounding data were acquired during the four intensive observation campaigns at Jungnang station. Among them, 148 soundings, which did not include the 23 precipitation cases, were used to estimate the ABLH (Table 1). It was observed that surface-based remote sensing instruments functioned poorly when the skies were cloudy. Thus, weather
conditions were divided into two categories, namely clear sky (cloud cover (CC) ≤ 30 %) and cloudy sky (CC ≥ 80 %).

**3.2 Ceilometer**

The backscattering coefficients observed by the ceilometer contain noise, especially near-range artifacts in the lower atmosphere near the lens of the instrument, atmospheric scattering due to strong daytime solar radiation, cloudiness, and precipitation. To reduce the noises in the raw data while maintaining the vertical and temporal characteristics of backscattering
coefficients, the temporal and spatial moving averages were applied. The moving average of the ceilometer backscattering coefficients are conducted by 10 range gates (100 m) and 10 time steps (10 minutes).

The SNR is introduced to prevent the noise from causing the estimation of ABLH at unreliable heights (de Haij et al., 2006; Heese et al., 2010; Kotthaus et al., 2016). Generally, backscattering coefficients at higher than the SNR stop level ($h_{SNR}$), which is the first altitude at which the SNR<1, are not used. The SNR at height $z$ is calculated by the formula put forward by
de Haij et al. (2007), as follows:

$$BN = \frac{1}{N}\sum_{z=12\,km}^{15\,km} \beta(z)\,, \tag{1}$$

$$\sigma_{\beta_{SNR}} = \sqrt{\frac{1}{N}\sum_{z=12\,km}^{15\,km}(\beta(z)-BN)^2}\,, \tag{2}$$

$$SNR(z) = \frac{\beta(z)}{BN+\sigma_{\beta_{SNR}}}\,, \tag{3}$$

where, $z$ is height, $\beta(z)$ is backscattering coefficient at $z$, $BN$ is background noise, which is calculated as the mean of $\beta(z)$
from 12 to 15 km, and $\sigma_{\beta_{SNR}}$ is the standard deviation of $\beta(z)$ at altitudes in excess of 12 to 15 km. When the upper layer contains a lot of noise, the SNR of the lower layer becomes smaller, and if the lower air is very clean, $h_{SNR}$ can be distributed in the lowest layer. Therefore, it is necessary to pay attention to the application taking into account the atmospheric condition





and lower aerosol concentration rather than using the SNR as an absolute reference. When calculating the SNR, $z$ was used from an altitude of 150 m to eliminate the discontinuity due to the limit of observation of surface-based remote sensing

instruments in the lower atmosphere.

Figure 2 shows the comparison of backscattering coefficients and $h_{SNR}$ before and after pre-processing. Before pre-processing (Fig. 2a), when the shortwave radiation was strong during the daytime, the noise was largely due to the scattering of sunlight and the SNR values were small. Especially, in the presence of daytime clouds (1400−1600 LST), the SNR became smaller and the $h_{SNR}$ was lowered. After pre-processing (Fig. 2b), the $h_{SNR}$ was stable even with strong solar radiation and daytime

clouds.

### 3.3 Microwave radiometer

The temperature and humidity from the microwave radiometer depends on the generalized atmospheric condition since they are estimated using an artificial neural network (Collaud Coen et al., 2014). Thus, it is difficult to detect fluctuations such as the upper air temperature inversion. Nevertheless, the stable-layer formed by the surface cooling at nighttime can be determined

only by the thermal parameter, and this study used the potential temperature calculated from the microwave radiometer for a limited height of less than 2 km and nighttime conditions to determine the SBLH.

The potential temperature can be computed by the vertical profiles of temperature, humidity, and pressure calculated using the ideal gas equation and assuming the hydrostatic equation. The vertical pressure $p_2$ at $z_2$ is calculated as:

$$p_2 = p_1 exp\left(-g\frac{z_2-z_1}{R\overline{T_z}}\right),\tag{4}$$

where, $p_2$ is the air pressure at height $z_2$ to be obtained, $p_1$ is the air pressure at $z_1$ below the $z_2$, $\overline{T_z}$ is the mean temperature between $z_1$ and $z_2$, $R$ is the gas constant for air (287 J kg$^{-1}$ K$^{-1}$), and $g$ is the gravitational acceleration. The potential temperature is calculated as:

$$\theta_z = T_z\left(\frac{p_0}{p_z}\right)^{\frac{R}{c_p}},\tag{5}$$

where, $\theta_z$ is the potential temperature at height $z$, $p_0$ and $p_z$ are air pressures at 1,000 hPa level and at height $z$, respectively.

$c_p$ is specific heat of dry air at constant pressure (1,004 J kg$^{-1}$ K$^{-1}$).

## 4 Methodology

### 4.1 Review of ABLH estimation method using radiosonde

The bulk Richardson number ($Ri_b$) is an approximation of the gradient Richardson number ($Ri$). $Ri_b$ is calculated as the ratio of the buoyancy term to the vertical wind shear using the vertical profiles of potential temperature and horizontal wind,

commonly defined as:

$$Ri_b = \frac{(g/\theta_0)(\theta_z-\theta_0)}{u_z{}^2+v_z{}^2}z,\tag{6}$$



where, $z$ is the height, $u_z$ and $v_z$ are the west-east and south-north wind speed, respectively, at $z$, $\theta_0$ is the surface potential temperature, and $\theta_z$ is the potential temperature at $z$. According to Stull (1988), laboratory research suggested that turbulence results when $Ri$ is smaller than the critical Richardson number, $Ri_c$. In many previous studies, they have experimented with

$Ri_c$ from 0.1 to 1.0 for the critical values (e.g., Holtslag and Boville, 1993; Jeričević and Grisogono, 2006; Esau and Zilitinkevich, 2010). Among them, 0.25 and 0.5 were most widely used (Zhang et al., 2014). In this study, the value of 0.5 is used as $Ri_c$ considering the small-scale fluctuation of the real observation value rather than the ideal profiles. Therefore, ABLH was determined to be the altitude at which $Ri_b$ would be less than 0.5.

As a method of determining ABLH during the nighttime, Collaud Coen et al. (2014) determined the nocturnal SBLH using

the temperature and potential temperature profiles from the radiosonde and microwave radiometer. Surface-based temperature inversion (SBI) represents the altitude at which the temperature decreases for the first time ($\Delta T/\Delta z = 0$), and this height can be regarded as SBLH (Stull, 1988; Seidel et al., 2010). When determining the nocturnal SBLH, it is possible to estimate the SBLH using the vertical profile of the thermal parameter only because the turbulence or aerosol layer characteristics can be used to detect the residual-layer at night. Therefore, the SBI method was used to estimate nocturnal SBLH using temperature

profiles from the radiosonde. In this study, the ABLH calculated using the radiosonde preferred to adopt the altitude determined by $Ri_b$, and the SBLH determined by the SBI method was replaced by the final ABLH when the nocturnal stable boundary-layer was formed.

### 4.2 Review of the ABLH estimation method using ceilometer

### 4.2.1 Time-variance method

The time-variance method (VAR) calculates the standard deviation ($\sigma_{\beta_{(z,t)}}$) of the backscattering coefficient profile measured by the ceilometer for 10 minutes using equation (4).

$$\sigma_{\beta_{VAR}} = \sqrt{\frac{1}{N}\sum_{t=1}^{N}\left(\beta(z,t) - \overline{\beta(z,t)}\right)^2} \, , \tag{7}$$

$$h_{VAR} = max(\sigma_{\beta_{VAR}}) \, , \ z < h_{SNR} \, , \tag{8}$$

where, $\beta(z,t)$ is the backscattering coefficient profile at time $t$, $\overline{\beta(z,t)}$ is the 10-minute mean backscattering coefficient, and

$N$ is the number of profiles ($N = 10$, in this study) . $\sigma_{\beta_{VAR}}$ represents the peak at high temporal variability, and thus, ABLH estimated by VAR ($h_{VAR}$) is determined as a height at which $\sigma_{\beta_{VAR}}$ shows a maximum and is less than $h_{SNR}$ (1,480 m). The $\sigma_{\beta_{VAR}}$ profile was smoothed to eliminate spurious variance peaks at small-scale fluctuations and above the $h_{SNR}$. Nevertheless, $\sigma_{\beta_{VAR}}\sigma_{\beta(z,t)}$ contains a spurious peak above the $h_{SNR}$ and tends to gradually increase with height. For the above reasons, the $h_{VAR}$ was calaulated only below the $h_{SNR}$.

Figure 3a shows the profiles of the $\sigma_{\beta_{VAR}}$ (red line), $\overline{\beta(z,t)}$ (black line), and $\beta(z,t)$ at one-minute intervals (dashed gray line) for 1050 to 1100 LST on 23 September 2016, and the ABLH determined by VAR ($h_{VAR} = 670$ m).



### 4.2.2 Gradient method

The gradient method is one of the most widely used methodologies for the estimation of ABLH. The largest negative peak of the first derivative with respect to the height of the backscattering coefficient from the ceilometer is determined as ABLH.

Generally, the first derivative (GM: gradient method), second derivative (IPM: Inflection Point Method), and logarithmic derivative (LGM: Logarithmic gradient method) are used, and the equations are as follows.

$$h_{GM} = min\left(\frac{\partial \beta(z)}{\partial z}\right), \tag{9}$$

$$h_{IPM} = min\left(\frac{\partial^2 \beta(z)}{\partial z^2}\right), \tag{10}$$

$$h_{LGM} = min\left(\frac{\partial \ln \beta(z)}{\partial z}\right), \tag{11}$$

Figure 3b shows the results of the gradient methods corresponding to 1100 LST on 23 September 2016. The bold solid line is a smoothed $\beta(z)$ profile, and the GM, IPM, and LGM results are represented by the solid, dotted, and dash-dotted line, respectively. The $h_{GM}$, $h_{IPM}$, and $h_{LGM}$ indicate ABLH with a maximum negative gradient for each method. The value of $h_{GM}$ (790 m) was slightly higher than $h_{IPM}$ (690 m) and lower than $h_{LGM}$ (1,580 m), which is consistent with Emeis et al. (2008). The second-largest negative (800 m) in LGM was similar to $h_{GM}$, and the second-largest negative in GM (1,570 m) was also

similar to the $h_{LGM}$ height. The $h_{IPM}$ is similar to $h_{VAR}$ (670 m), and both are located at the altitude where $\beta(z)$ begins to decrease sharply. The altitude at which the largest negative gradient appears for each derivation method is different, but it appears at altitudes similar to the second peaks.

### 4.2.3 Wavelet covariance transform method

The wavelet covariance transform method is also one of the most widely used methods (WAV). WAV uses the Haar step

function, which is defined as:

$$h\left(\frac{z-b}{a}\right) = \begin{cases} +1: & b - \frac{a}{2} \le z \le b \\ -1: & b \le z \le b + \frac{a}{2} \\ 0: & elsewhere \end{cases}, \tag{12}$$

where, $b$ is translation of function (the location at which the function is centered), and $a$ is dilation of function (the spatial extent). The covariance transform of Haar function, $W_\beta$, is defined as,

$$W_\beta(a,b) = \frac{1}{a}\int_{z_b}^{z_t} \beta(z)h\left(\frac{z-b}{a}\right)dz, \tag{13}$$

$$h_{WAV} = max(W_\beta(a,b)), \tag{14}$$

where, $z_b$ and $z_t$ are the bottom and top heights of the profile. The altitude with the maximum value of $W_\beta(a,b)$ is determined by ABLH ($h_{WAV}$). In this study, $a$ is set to 24 dilations at 15 m interval from 15 m to 360 m, and $b$ is set in 10 m step size from 60 m to 3,000 m, respectively.



Davis et al. (2000) illustrated the importance of determining the dilation through experiments that use the airborne lidar
backscattering profile. Smaller dilations are sensitive to small-scale fluctuations of $\beta(z)$ and include noise, while larger
dilations tend to ignore small-scale structures and detect changes in scale such as the entrainment zone. Especially in the real
atmosphere, small-scale fluctuation of $\beta(z)$ due to sudden turbulence appears, and it plays an important role in mechanical
mixing in ML. In order to consider small-scale features, $W_\beta(a,b)$ profiles were processed by averaging over $a < 100$ m
(WAV1), $a > 300$ m (WAV2), and the total $a$ (WAV3) (de Haij et al., 2007). The height with the maximum values of $W_\beta(a,b)$
by WAV1, WAV2, and WAV3 can be determined as ABLH ($h_{WAV1}$, $h_{WAV2}$, $h_{WAV3}$), respectively.

Figure 3c shows the results of the wavelet method. The bold solid line is smoothed $\beta(z)$ and the solid line, dashed line, and
dash-dotted line indicate the results of WAV1, WAV2, and WAV3, respectively. As described in 4.2.2 above, $\beta(z)$ decreases
rapidly at altitudes of approximately 700 m and 1,500 m, and $W_\beta(a,b)$ also peaks at very close altitudes. In WAV1, the first
peak ($h_{WAV1}$) appeared at 680 m, which is very close to $h_{VAR}$ (670 m) and $h_{IMP}$ (690 m). WAV2 (WAV3) showed two peaks
at 750 m (730 m) and 1,550 m (1,550 m). The first peaks ($h_{WAV2}$, $h_{WAV3}$) were similar to $h_{GM}$ (790 m) and the second peaks
were similar to $h_{LGM}$ (1,580 m; also second peak of the $h_{GM}$).

### 4.2.4 Clustering analysis method

The k-means clustering analysis (CLST) is a nonhierarchical clustering method that can determine the ABLH by dividing the
distribution where the backscattering coefficient profile from the ceilometer sharply decreases or increases. The cluster center
is determined as the point at which the sum of the squared errors (Toledo et al., 2014) is minimized. The number of cluster
seeds is determined by the Dunn index (Dunn, 1974; Toledo et al., 2014).

Figure 3d shows the results of the ABLH calculation using k-means clustering analysis at 1100 LST on 23 September 2016.
As a result of the cluster validation, the optimal number calculated by the Dunn index was 3, and the cluster were distinguished
at 800 m ($h_{CLST1}$) and 1,430 m ($h_{CLST2}$). The altitude at which a cluster changes to another cluster can be determined as ABLH.
The value of $h_{CLST1}$ was similar to $h_{GM}$ (790 m) and $h_{WAV1}$ (770 m). $h_{CLST2}$ was slightly lower than $h_{LGM}$ (1,580 m) and $h_{WAV2}$
(1,530 m).

### 4.3 Nocturnal SBLH estimation using microwave radiometer

It is possible to estimate the nocturnal SBLH by determining the thermal stability and instability from the microwave
radiometer-derived vertical profiles of thermal parameters such as temperature and potential temperature (Collaud Coen et al.,
2014; Saeed et al., 2016). Given the vertical profile the atmospheric temperature, it is possible to determine the altitude of
$dT/dz = 0$ according to the SBI method to determine the thermal stability. However, in real atmospheric conditions, the air
parcel follows the environmental lapse rate (ELR), which differs depending on the time and place rather than the theoretical
lapse rate (TLR), and the criterion of the potential temperature gradient is also dominant in the ELR. In this study, it is assumed





that there is a high possibility that SBL exists when $d\theta/dz$ near the surface is larger than the ELR. After that, we set the

threshold ($\overline{\Gamma_f}$) of the ELR considering the vertical variability of $d\theta/dz$ to distinguish the distinct layers.

Figures 4a−b show the vertical profiles of the potential temperature and the vertical gradient of the potential temperature obtained by a microwave radiometer at Jungnang station at 1500 LST (solid line), 2100 LST (dashed line) on 23, and 0000 LST (dotted line) on 24 September 2016. The potential temperature decreases with height at a constant rate above 2,000 m (Fig. 4a), and it can be considered a slope of the ELR. The TLR and ELR are shown in Fig. 4b as solid and dashed gray lines,

respectively. It was thermally unstable at 1500 LST on 23 when the value near the surface was smaller than the TLR (Fig. 4b). As the near-surface temperature decreased due to surface cooling after sunset, and a stable layer with a positive value of $d\theta/dz$ appeared, the slope of $d\theta/dz$ increased and a more stable layer was formed at 0000 LST on 24. At this time, the daily mean potential temperature gradient in free atmosphere over 2,000 m was 5.5 K km$^{-1}$, and this values is used as the threshold $\overline{\Gamma_f}$ for the ELR.

Thus, it can be concluded that the layer is considered as a stably affecting layer if the $d\theta/dz$ is larger than $\overline{\Gamma_f}$ and unstably affecting if the $d\theta/dz$ is smaller. The $d\theta/dz$ in the lower atmosphere at 2100 LST on 23 is larger than 0 K km$^{-1}$, which is the stable condition in the TLR criterion, but is smaller than 5.5 K km$^{-1}$, and so, it is difficult to determine it as stable in the ELR. Figure 4c shows the vertical variance of $d\theta/dz$. The vertical variance was calculated for 150 m at each altitude. At 1500 LST on 23, which was well mixed vertically, the variance of $d\theta/dz$ near the surface was close to 0 K km$^{-1}$, whereas there was

large variance of $d\theta/dz$ at 2100 LST on 23 and 0000 LST on 24. It is possible to determine the altitude at which the vertical variance decreases rapidly (500 m; gray line at Fig. 4b) at 0000 LST on 24, satisfying the ELR condition, and $d\theta/dz$ at the altitude is 3.6 K km$^{-1}$.

Since the $\overline{\Gamma_f}$ and $d\theta/dz$ depend on the time, we find the altitude at which the vertical variance of the daily data decreases sharply every 10 minutes while satisfying the stable ELR condition ($> \overline{\Gamma_f}$) for threshold setting. For the classification of the

distinct layer, find the altitude of the maximum vertical variance during a day and determine the potential temperature gradient of that day as the critical lapse rate (CLR; $\Gamma_{cr}$) of that day.

$$Var\left(\frac{\partial\theta}{\partial z}\right)_z = \frac{1}{H}\sum_{z=1}^{H}\left[\left(\frac{\partial\theta}{\partial z}\right)_z - \left(\frac{\overline{\partial\theta}}{\partial z}\right)_z\right]^2 , \tag{15}$$

$$\Gamma_{cr} = \max_{t=1day}\left\{\max\left(Var\left(\frac{\partial\theta}{\partial z}\right)\right)_t\right\} , \tag{16}$$

where, $Var\left(\frac{\partial\theta}{\partial z}\right)_z$ is the vertical variance of potential temperature gradient at z height, $\left(\frac{\partial\theta}{\partial z}\right)_z$ is potential temperature gradient

at z height, $\left(\frac{\overline{\partial\theta}}{\partial z}\right)_z$ is mean potential temperature gradient over ±150 m at z height, and $H$ is number of the vertical interval ($H = 6$; 300 m).



As a result, on 23 September, $\Gamma_{cr}$ was 7.0 K km$^{-1}$, and the altitude at which the $d\theta/dz$ profile crosses CLR was determined as SBLH. In order to improve the quality of microwave radiometer data, if there are surface heating by shortwave radiation (net radiation > 0 W m$^{-2}$) and precipitation, it is removed.

### 4.4 Integrated system for ABLH estimation

In a real atmosphere, there is not only one ABL, but a complex structure with several layers that depend on time, place, and atmospheric phenomena. Therefore, ABLH shows large differences among methodologies and is an arbitrary decision by the researcher. In this study, an integrated system for ABLH estimation (ISABLE) was developed to determine the optimal ABLH. ISABLE applies the four methodologies described above using the backscattering coefficient from the ceilometer, along with the CLR method that uses the potential temperature profile from microwave radiometer.

#### 4.4.1 Integration method

Figure 5 shows the schematic flow of the ABLH candidate group selection process. The INPUT is the ABLHs estimated by applying the four methods using a backscattering coefficient from the ceilometer, the present study was estimated up to 19 layers. The VAR selects a maximum of 3 peaks as the ABLH candidates. In the GM, a maximum of five peaks are found to reduce redundancy. In the WAV method, up to three altitudes are selected as ABLH candidates for WAV3 considering the entire dilation, and WAV1 and WAV2 are selected as ABLH candidates for two altitudes in order to minimize the redundancy with WAV3. The CLST was limited to four altitudes to avoid taking the noise structure into account.

The ABLH candidate groups are selected by k-means clustering analysis for a maximum of 19 ABLHs. When selecting the ABLH candidate groups, a gap of 150 m is placed between each ABLH candidate to prevent overlap at close altitude. As a result of the first clustering, groups with three or more members and RMSE less than or equal to 50 m are classified into the ABLH candidate groups. In the first clustering result, if the number of members is less than 3 and the RMSE is greater than 50 m, it is excluded from the ABLH candidate groups. If the number of members is larger than or equal to three, but the RMSE exceeds 50 m, a second clustering analysis is performed.

The results of the second clustering analysis are classified as the ABLH candidate groups of clusters with number of members greater than or equal to 2 and the RMSE when the height is less than or equal to 50 m, while the member is removed if the number of members is less than 2. If the second clustering result is larger or equal than 2 cluster members but the RMSE exceeds 50 m, the member with the largest distance from the average of that cluster is removed and re-evaluated. The above procedure is repeated until the number of members of the cluster is larger than or equal to 2 and the RMSE satisfies 50 m or less. These are then classified as ABLH candidate groups.

The final OUTPUT, the ABLH candidate groups, is ranked in descending order of the number of members, and if the number of members is the same, the RMSE is ranked in ascending order. The five groups are selected, and the average of each group is determined as the final ABLH estimated by the ceilometer backscattering coefficient.





### 4.4.2 ISABLE post-processing

Various ABLH estimation algorithms are merged through ISABLE. However, there are still limits to estimating the ABLH

such as observational errors and small-scale fluctuations in real atmosphere, and appropriate post-processing is required (Kotthaus and Grimmond, 2018). Unreasonable ABLHs such as the ABLH above $h_{SNR}$, caused by instrument-related near-range artifacts, and the isolated ABLH related small-scale structure are removed through the three-step post-process.

Figure 6a shows the ABLHs determined by ceilometer observations without post-processing (CM_ABLH) from 1800 LST on 22 to 1200 LST on 25, September 2016. There are not only ABLHs at higher than $h_{SNR}$ at 1000−1200 LST on 25, but near-

range ABLHs in the daytime (1200−1600 LST), when the convective is developed well, and isolated ABLHs that seem independent without time-space continuity are formed. First, the ABLH is removed if higher than $h_{SNR}$, and as a result, the ABLHs which appeared at approximately 2,500 m at 1000-1200 LST on 25, was removed (Fig. 6b). As previously described (section 3.2), the altitude higher than $h_{SNR}$ contained less meaningful information because the backscatter signal compared to the background noise is weak. Thus, the ABLH above the $h_{SNR}$ is removed. Second, the lower layer ABLHs during daytime,

which is represented by the near-range artifacts, was removed (Fig. 6c). The ABLH grows slowly after sunrise, while it grows rapidly approximately 1−2 h before noon. The maximum ABLH appears approximately 2−3 h after noon (1400−1600 LST). During this time period, the near-surface is heated and atmospheric mixing by convection is active, and thus ABLH grows to the maximum. Therefore, the ABLH that appears in the lower layer at this time can be regarded as inappropriate. Using the radiation observation at Jungnang station, the convective mixing period is set from 1 h before the time of maximum net

radiation to 1 h after sunset (the net radiation is 0 W m$^{-2}$), and ABLHs below 500 m at that time are unreasonable ABLHs that arise from instrument-related near-range artifacts (Fig. 6b). Third, in order to find the discontinuous ABLH caused by small-scale fluctuations and a separated small-scale aerosol layer, the ABLH is considered to be discontinuous if no other ABLHs are present within ± 10 time step (100 minutes) and ± 12 range gates (120 m). Also, the density-based spatial clustering of applications with noise (DBSCAN; Ester *et al.*, 1996) method may be used to eliminate the isolated ABLH. DBSCAN is an

algorithm that extracts the noise contained in a cluster. Each point (core point) of a cluster and neighborhoods (border points) within a given radius (ε) must contain a minimum number of points (MinPts) within that ε. To apply the same ε to the time-height axes, DBSCAN is performed on a normalized ABLH with values between 0 and 1 for both axes per day. Figure 6d shows the result discontinuity check and DBSCAN with $\varepsilon = 0.0125$ ($t = 72$ mins; $z = 56\ m$) and MinPts = 3. The ABLHs that appeared independently were removed, and the boundary layer distinction became more pronounced.

Figure 6e shows the backscattering coefficient and CM_ABLH from those after post-processing. In addition, the nocturnal SBLH estimated by the microwave radiometer (MWR_ABLH) is merged and determines the lowest of them as the final ABLH determined by ISABLE (ISABLE_ABLH).



## 5 Results

### 5.1 Diurnal variation of ABLH from radiosonde

ABLHs were calculated for 148 radiosonde observations made at the Jungnang station in Seoul during the year from 2015 to 2018. Figure 7 shows the diurnal variation of the ABLH. The maximum mean ABLH estimated by radiosonde was 1,019 m (median = 925 m) at 1500 LST and the minimum mean height was 418 m (median = 250 m) at 0600 LST. The SBL is developed by surface cooling when the air is clean and long wave radiation is strong during night, but this is not always possible. As can be seen in Fig. 7 ABLH of more than 1 km altitude appeared as outliers at nighttime (2100−0900 LST). The ABLH is observed

in the upper layer at nighttime if the ML is developed due to the nocturnal heat island phenomenon at Jungnang station located in the urban area. The interquartile range (IQR; Q3 − Q1) showed the minimum value at 0900 LST (268 m) and was maximum at 1800 LST (740 m). In general, at nighttime, ABLHs are concentrated in the lower layer, and IQR values increase as the ML develops after sunrise.

### 5.2 ISABLE performance assessment

Figure 8 shows the ABLHs obtained by radiosonde observation (RS) and the ISABLE, as well as the results of each methodology obtained using a ceilometer and a microwave radiometer from 1800 LST on 22 September 2016 to 1200 LST on 25 September 2016. The ABLH estimated from radiosonde showed the maximum height at 1500 LST during the daytime and low height during the nighttime. The same diurnal variation can be seen in the ISABLE results. The correlation coefficient R between radiosonde and ISABLE showed a high correlation of 0.98, with was an average bias of -101 m, and RMSE was

found to be 135 m. Other methodologies showed maximum height, similar to the radiosonde during the daytime, but often, ABLH still appears in the upper levels at nighttime. This is considered to have been the ABLH estimated at the altitude where a stronger signal appears in the residual layer of the upper layer at nighttime. ISABLE complemented the above shortcomings by integrating the four methodologies and also considering the stable layer that appears at night.

Table 2 shows the results of verifying ISABLE as radiosonde by separating the entire period (N=148), the daytime (N=67; 0900−1800 LST), and the nighttime (N=64; 2100−0600 LST). Not only the ABLH of ISABLE but also the results estimated

by different methodologies (VAR, GM, WAV1, WAV2, WAV3, and CLST) are compared. The correlation coefficient of ISABLE and radiosonde for the entire period was 0.72, mean bias was −34 m, and RMSE was 322 m. By this methodology, the correlation coefficients are high and the RMSE is low in the order of VAR and CLST methods. Even during the daytime, the correlation between ISABLE and radiosonde was the highest (0.90), and the mean bias and RMSE was the lowest (−9 m and 203 m, respectively). The VAR (R=0.72, Bias=17 m, RMSE=308 m) results was better after ISABLE and CLST (R=0.68,

Bias=−20 m, RMSE=369 m) was the next. At the nighttime, VAR was the best results (R=0.46, Bias=333 m, RMSE=403 m), ISABLE showed R of 0.37, Bias of −43 m, and RMSE of 394 m, which shown that the correlation is not good, but Bias and RMSE showed the best verification results. Although the ISABLE has shown improved performance over the entire period and during the daytime, compared to other methodologies, there are still limitations in estimating ABLH at nighttime.





Verification score of the ABLHs estimated using each of the methodologies was lower than that obtained by the ISABLE. There were 28 cases where the difference between the ABLH estimated using each methodology and the RS_ABLH was greater than 1,000 m. Of the 28 cases, 20 showed strong backscattering coefficients in the aerosol residual layer at night; ABLH was estimated at the corresponding altitude, especially using the GM and WAV2 methods. The remaining 8 cases occurred during the daytime, of which 6 were in the presence of clouds, and 2 were in the case of very clear sky with weak

backscattering signal. The remaining 2 cases were related to the SBL, for which SBLH could not be obtained despite using radiosonde. The above cases often appear in real atmosphere; however, it is difficult to estimate consistent ABLH under the aforementioned atmospheric condition.

The correlation coefficients and RMSEs of ABLH obtained by ISABLE were higher than those obtained using previous methodologies, especially GM and WAV2. In the case of clear sky (N=73), ISABLE exhibited the highest R (0.81), followed

by WAV2 (0.79) and GM (0.77). The mean bias and RMSE of ABLH estimated by ISABLE (Bias=−99 m, RMSE=286 m) were slightly lower than those by the GM (Bias=195 m, RMSE=308 m) and WAV2 (Bias=170 m, RMSE=296 m) methods. In the case of cloudy sky (N=15), the performances of VAR (R=0.64, Bias=139 m, RMSE=285 m) and CLST (R=0.60, Bias=58 m, RMSE=287 m) were relatively good. The ABLH estimated by ISABLE (R=0.59, Bias=58 m, RMSE=287 m) also revealed a good performance compared to that by VAR and CLST.

The correlation coefficient of ABLH obtained by ISABLE for the entire daytime (N=59) showed a high value of 0.83. In comparison, WAV2, GM, and VAR showed R of 0.85, 0.81, and 0.80, respectively. The mean bias and RMSE of ISABLE also demonstrated a high score of −59 m and 0.84, respectively, for clear sky during daytime (N=41). WAV2 revealed the following score: R=0.89, Bias=72 m, and RMSE=206 m; GM showed the following scores: R=0.82, Bias=115 m, and RMSE=258 m. In the case of cloudy sky (N=7), ISABLE showed the best performance, that is R=0.84, Bias=109 m, and

RMSE=168 m; this was followed by CLST having R=0.84, Bias=101 m, and RMSE=178 m. For the entire nighttime (N=44), the correlation coefficients of GM and WAV2 were 0.62 and 0.59, respectively, while that of ISABLE was 0.52. However, the mean bias of ABLH obtained by GM and WAV2 was 326 m and 335 m, which was smaller than that (−35 m) obtained by ISABLE. In case of clear sky (N=22), the correlation coefficient of ABLH obtained by GM was the highest (0.77), and that by WAV2 was the second highest (0.75). Moreover, ISABLE showed a correlation coefficient of 0.66 and a mean bias of

−124 m, which was much lower than that obtained using GM (343 m) and WAV2 (346 m). The RMSE of ABLH obtained by ISABLE was 351 m, which was slightly higher than that using GM (301 m) and WAV2 (306 m). In the case of cloudy sky (N=8), VAR exhibited the best performance (R=0.55, Bias=78 m, and RMSE=344); this was followed by ISABLE (R=0.52, Bias=46 m, RMSE=398 m). Overall, most ABLHs were higher during the day and under clear sky conditions. Furthermore, ABLH obtained using ISABLE was higher than the previous methodologies both under clear and cloudy skies as well as during

daytime and nighttime (Fig. 9).

Not many SBLs were observed at the Jungnang station due to the effect of urban heat island in the urban areas (Park et al., 2014). The nocturnal SBL observed by radiosonde was merely 3 (5) times higher than those by microwave radiometer during





the verification (entire) period. Although the number of observations were not sufficient to verify the SBLH, MWR_SBLH was similar to RS_SBLH, having a mean deviation of 8.4 m and RMSE of 75 m.

### 5.3. Diurnal and seasonal variation of ABLH from ISABLE

For the period from November 2015 to October 2018, the ISABLE ABLH was determined by vertical profiles of backscattering coefficient from the ceilometer and potential temperature from microwave radiometer at Jungnang station in Seoul. Figure 10 shows the diurnal variations over the observation period of clear (Fig. 10a) and cloudy (Fig. 10b) skies. The maximum mean ABLH on the clear skies was 1,202 m at 1600 LST, while it was 1,085 m at 1500 LST on cloudy skies. The mean ABLHs of clear skies were higher than that of cloudy skies during the daytime, but, there was no significant difference at nighttime. Generally, IQR values of ABLH were large during the daytime and small at the nighttime, particularly, when ML was developing (1100−1200 LST) and declining or SBL was forming (1800−1900 LST), or during the transition (Fig. 10a).

Figure 11 shows the diurnal variations of ABLH for clear skies by season. The maximum seasonal mean ABLH was 1,268 m in MAM (March, April, May; Fig. 11b) at 1500 LST and the second highest was 1,256 m in JJA (June, July, August; Fig. 11c) at 1600 LST. DJF (December, January, February; Fig. 11a) shows the lowest seasonal mean ABLH at 1,092 m at 1600 LST. This is consistent with the results of the radiation analysis at an urban residential area in Seoul (Park et al., 2014). The lowest seasonal mean ABLH was 341 m at 0200 LST in DJF, and it was formed relatively high at 462 m at 0600 LST in JJA. The nighttime in the JJA period indicates less thermodynamically stable conditions than DJF, and because of the urban characteristics of the observatory, it is possible to form an ML in the lower layer even at nighttime, as a result, ABLH can be estimated higher than DJF at nighttime.

Seasonal IQR is small at the nighttime and large at the daytime in all seasons. The difference in IQR values between the daytime and the nighttime by season was obvious in DJF, MAM, and SON, but not in JJA. In MAM, the difference between the maximum and minimum of IQR was 781 m, while during JJA, it was only 428 m. In comparison to other seasons, this means that the ABLH is calculated within a relatively uniform statistical range, regardless of the time in JJA.

The ML and SBL growth and decline are affected by changes in sunrise and sunset times depending on the season. In this transition period, the uncertainty increases in the ABLH estimation and the IQR value is increased. The peaks of IQR appeared at 1200 LST and 1800 LST in DJF, and 1100 LST and 1900 LST in MAM, it can be seen that the development of the ML was quickened and the decline of the ML and the SBL formation were delayed. In JJA, IQR was large at 1000 LST and 2000 LST, this shows that the ML was the earliest developed and the latest declined. In SON (Fig. 11d), large values of IQR appeared again at 1200 LST and 1800 LST that shows the sunrise was delayed and sunset was quickened.

Figure 12 shows the seasonal distribution of ABLH of during the daytime (1400−1600 LST) and the nighttime (0300−0500 LST). The mean ABLH of the daytime (Fig. 12a) was similarly high at 1,240 m in MAM and 1,230 m in JJA, but IQR was larger than MAM (440 m) in JJA (520 m). In DJF, the mean ABLH was the lowest (1,060 m) and the IQR was the smallest at 330 m. At the nighttime (Fig. 12b), ABLH was the highest at 480 m in JJA, and IQR also was the largest at 290 m. The mean ABLH of the other seasons were MAM 417 m, SON 370 m, DJF 350 m, and IQR were 170 m, 140 m, and 130 m, respectively.



The mean ABLH in MAM and JJA were relatively high in both the daytime and the nighttime, and IQR was the largest in summer.

# 6 Summary and discussion

In this study, ISABLE, a method for optimized ABLH estimation, was developed by applying statistical techniques to ABLHs

determined by conventional estimation methodologies. A maximum of five ABLHs were estimated every 10 minutes using the ceilometer backscattering coefficient for each methodology (time-variance method, gradient method, wavelet covariance transform method, clustering analysis method). The determined ABLHs were divided into five clusters by k-means cluster analysis, and the ABLH was finally determined as the average of the members of the clusters satisfying the statistical conditions. The nocturnal SBLH was estimated using a potential temperature profile from the microwave radiometer. The SBLH was

determined using the CLR method, which uses the threshold of an environmental lapse rate of potential temperature over a day. The ABLHs estimated by the ceilometer were post-processed in three steps (SNR threshold, instrument-related near-range artifact, isolated ABLHs) to remove unreasonable values. The lowest altitude among the ABLH and the nocturnal SBLH was finally determined as an optimized ISABLE ABLH.

From 2015 to 2018, ABLHs were determined using the ISABLE (ISABLE_ABLH) at 10-minute intervals, and compared to

and verified with the ABLH estimated by radiosonde observations (RS_ABLH) at Jungnang station in Seoul City, Korea. The $Ri_b$ was calculated using the vertical profile of potential temperature and wind obtained by radiosonde observation to estimate the ABLH by radiosonde during the entire day, and using the SBI method at the nighttime, the nocturnal SBLH was determined by the vertical temperature profile. The performance of ISABLE was verified by comparing the ISABLE_ABLH as well as the ABLH estimated from each methodology. A total of 148 radiosonde observations were used in the verification, and the

correlation between ISABLE_ABLH and RS_ABLH was the highest compared to other methodologies with R values of 0.72. The mean bias and RMSE were the lowest with −34 m and 322 m respectively, resulting in the best verification results. In addition, it was verified by separating the daytime (0900−1800 LST) and nighttime (2100−0600 LST). As a result, the correlation between ISABLE_ABLH and RS_ABLH was the highest at 0.90, and the mean bias and RMSE were the lowest at −9 m, and 203 m respectively. On the other hand, during the nighttime, the result of the VAR method had the highest $R^2$ at

0.21, but the mean bias and RMSE were the lowest at ISABLE_ABLH at −43 m, 394 m, respectively. Generally, the poor performance was due to multiple factors, such as strong backscattering signals in the residual layer, presence of clouds, and weak backscattering signal. When these factors were excluded, the performance for the remaining 118 cases was much improved. In addition, the effect of weather conditions on the performances of ABLH estimation was evaluated by classifying the cases into clear and cloudy skies. It was observed that the performance of ABLH estimated by GM and WAV2 was better

in the case of clear skies. Furthermore, ABLH estimated by GM and WAV was significantly poor, whereas ABLH calculated using VAR and CLST was found to be better for cloudy skies. Thus, the performance of ABLH by ISABLE showed the best results in most categories. Based on the above results, the results of ABLH estimations can be improved by using ISABLE



rather than using a single methodology. The ABLHs estimated by ISABLE are expected to have great potential in parameterizing vertical diffusion in the ABL and to understand severe haze/smog events fumigated from the upper layer during
the ABL evolution (typically on 0900−1200 LST).

The diurnal variation of the ISABLE was also analyzed for the period from November 2015 to October 2018. ABLH began to grow at 0900 LST to 1100 LST after sunrise, and after sunset, from 1800 LST to 2000 LST, the ML was lowered and the SBL was formed, indicating diurnal variation. At this time, if the SBL is present according to the atmospheric condition, it is determined as ABLH, but otherwise, top of the residual-layer is determined as ABLH, so the IQR of ABLH was larger than
other times. The features of a clear day and cloudy day can be identified by the deviation of the IQR during the daytime and nighttime. The IQR was large during daytime and low at the nighttime, and the deviation of the daytime and nighttime was greater on clear days. The seasonal features were high altitudes of maximum ABLH in MAM and JJA, while the minimum ABLH values were the lowest in DJF. The IQR showed significant differences between the daytime and nighttime in DJF, MAM, and SON, and relatively small values in JJA. In addition, depending on the seasonal sunrise and sunset time fluctuations,
DJF showed two peaks at 1000 LST and 1800 LST, and the peaks at 0900 LST and 2000 LST in JJA.

Although the studies on ABLH estimation using surface-based remote sensing instrument such as ceilometers and microwave radiometers have been actively pursued, ABLHs may differ depending on the estimation methodologies. Although ISABLE-estimated ABLH exhibited better performance than those estimated by the earlier conventional methodologies, there are still limitations. In particular, ABLHs estimated from ceilometer in the lower layer (up to 500 m) are not reliable due to near-range
artifacts. This issue could be supplemented by the temperature profile obtained by a microwave radiometer only at the nighttime. ABLHs are difficult to estimate under cloudy sky or precipitation, severe fog and smog events. These limitations and drawbacks should be overcome by combining several observation equipment.

*Acknowledgements* This work was funded by the Korea Meteorological Administration Research and Development Program
under Grant KMI2018-05310. Most data used in this study were supported by the Korea Meteorological Administration's National Institute of Meteorological Sciences Development of Advanced Research on Biometeorology and Industrial Meteorology (136500304) and the Hankuk University of Foreign Studies.

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

**Table 1: Information of GPS radiosonde observations at Jungnang station in Seoul City, Korea.**

| Observation period | Interval (hour) | Number of observations | Number of precipitation | |
|---|---|---|---|---|
| | | | Yes | No |
| **2015.11.23. ~ 11.30** | 3 | 54 | 10 | 44 |
| **2016.09.19. ~ 09.30** | 3 | 60 | 6 | 54 |
| **2016.10.02. ~ 10.07** | 6 | 29 | 7 | 22 |
| **2018.09.10. ~ 09.17** | 6 | 28 | 0 | 28 |
| | | 171 | 23 | 148 |

**Table 2: Statistical performance between ABLHs obtained by various methods including ISABLE and radiosonde observations for**
**all data (N=148), daytime (N=67; 0900 to 1800 LST), and nighttime (N=64; 2100 to 0600 LST).**

| | Method/Score | VAR | GM | WAV1 | WAV2 | WAV3 | CLST | ISABLE |
|---|---|---|---|---|---|---|---|---|
| **All (148)** | R | 0.60 | 0.41 | 0.17 | 0.41 | 0.26 | 0.45 | 0.72 |
| | Bias (m) | 219 | 420 | 187 | 414 | 289 | 125 | **−34** |
| | RMSE (m) | 372 | 519 | 631 | 537 | 585 | 474 | **322** |
| **Day (67)** | R | 0.72 | 0.59 | 0.17 | 0.50 | 0.22 | 0.68 | 0.90 |
| | Bias (m) | 17 | 275 | 38 | 288 | 162 | −20 | **−9** |
| | RMSE (m) | 308 | 431 | 665 | 509 | 619 | 369 | **203** |
| **Night (64)** | R | 0.46 | 0.33 | 0.22 | 0.37 | 0.26 | 0.22 | 0.37 |
| | Bias (m) | 333 | 586 | 331 | 561 | 449 | 296 | −43 |
| | RMSE (m) | 403 | 543 | 575 | 524 | 565 | 549 | 394 |



**Table 3: Statistical performance between ABLHs obtained by various methods including ISABLE and radiosonde observations for 118 cases having no major error factors. Each case was divided into clear and cloudy skies.**

|  | Method/Score | VAR | GM | WAV1 | WAV2 | WAV3 | CLST | ISABLE |
|---|---|---|---|---|---|---|---|---|
| **All (118)** | R | 0.68 | 0.74 | 0.39 | 0.74 | 0.51 | 0.71 | 0.79 |
|  | Bias (m) | 164 | 277 | 23 | 217 | 121 | 13 | −37 |
|  | RMSE (m) | 333 | 306 | 461 | 304 | 409 | 321 | 281 |
| **CC ≤ 30%** | R | 0.70 | 0.77 | 0.32 | 0.79 | 0.50 | 0.70 | 0.81 |
| **(73)** | Bias (m) | 125 | 195 | −42 | 170 | 65 | −38 | −99 |
|  | RMSE (m) | 344 | 308 | 509 | 296 | 435 | 348 | 286 |
| **CC ≥ 80%** | R | 0.64 | 0.46 | 0.46 | 0.45 | 0.45 | 0.60 | 0.59 |
| **(15)** | Bias (m) | 139 | 239 | 162 | 249 | 219 | 58 | 70 |
|  | RMSE (m) | 285 | 349 | 357 | 351 | 359 | 287 | 320 |
| **Daytime (59)** | R | 0.80 | 0.81 | 0.39 | 0.85 | 0.54 | 0.77 | 0.91 |
|  | Bias (m) | 72 | 165 | −83 | 133 | 32 | −59 | −22 |
|  | RMSE (m) | 270 | 263 | 472 | 237 | 408 | 299 | 191 |
| **CC ≤ 30%** | R | 0.80 | 0.82 | 0.32 | 0.89 | 0.55 | 0.77 | 0.92 |
| **(41)** | Bias (m) | 26 | 115 | −151 | 72 | −28 | −137 | −59 |
|  | RMSE (m) | 266 | 258 | 493 | 206 | 399 | 301 | 181 |
| **CC ≥ 80%** | R | 0.83 | 0.62 | 0.64 | 0.62 | 0.57 | 0.84 | 0.84 |
| **(7)** | Bias (m) | 209 | 287 | 173 | 293 | 251 | 101 | 109 |
|  | RMSE (m) | 201 | 275 | 294 | 276 | 307 | 178 | 168 |
| **Nighttime (44)** | R | 0.45 | 0.62 | 0.26 | 0.59 | 0.32 | 0.54 | 0.52 |
|  | Bias (m) | 271 | 319 | 127 | 335 | 220 | 110 | -35 |
|  | RMSE (m) | 376 | 326 | 439 | 334 | 412 | 335 | 328 |
| **CC ≤ 30%** | R | 0.54 | 0.77 | 0.17 | 0.75 | 0.28 | 0.60 | 0.66 |
| **(22)** | Bias (m) | 293 | 343 | 120 | 346 | 195 | 145 | -124 |
|  | RMSE (m) | 397 | 301 | 533 | 306 | 484 | 371 | 351 |
| **CC ≥ 80%** | R | 0.55 | 0.37 | 0.35 | 0.35 | 0.37 | 0.48 | 0.52 |
| **(8)** | Bias (m) | 78 | 198 | 152 | 211 | 191 | 20 | 46 |
|  | RMSE (m) | 344 | 418 | 425 | 421 | 418 | 365 | 398 |




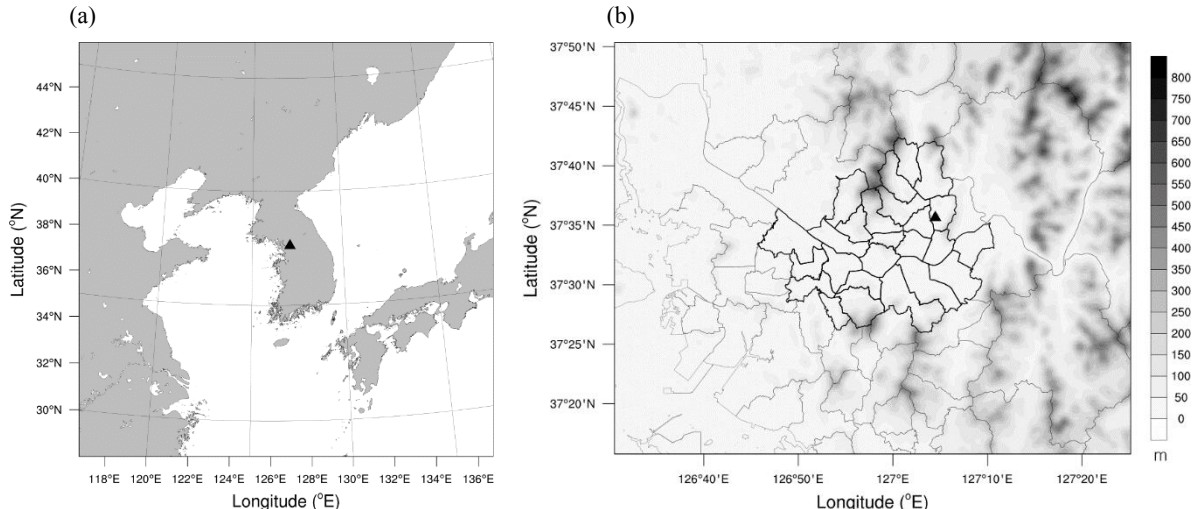

Figure 1: Location of Jungnang station (triangle) in (a) East Asia, (b) the Seoul Metropolitan Area with its topography.





**Figure 2: Time−height cross sections of the backscattering coefficient obtained by a ceilometer and signal-to-noise ratio (SNR) stop**
**level ($h_{SNR}$) (a) before and (b) after pre-processing**




**Figure 3: Examples of ABLH estimations: (a) Time-variance method. 10-min averaged $\bar{\beta}_{(z)}$ (bold black line) and standard deviation ($\sigma_{\beta_{(z,t)}}$) (red line) at 1100 LST on 23 September 23 2016, the gray curves are $\beta_{(z)}$ at 1 minute intervals from 1050 LST to 1100 LST; $h_{VAR}$ is the ABLH retrived by the time-variance method; (b) Gradient method (GM, IPM, LGM). bold black line indicates 10-min averaged $\bar{\beta}_{(z)}$, thin solid, dashed, and dash-dotted lines indicate results of GM, IPM, and LGM, respectively. the ABLH ($h_{GM}$, $h_{IMP}$, $h_{IPM}$) determined by each method; (c) Wavelet covaraince transform method (WAV). Bold black line indicates 10-min averaged $\bar{\beta}_{(z)}$, thin solid, dashed, and dash-dotted line indicate results of WAV1, 2, and 3, respectively; and $h_{WAV1}$, $h_{WAV2}$, and $h_{WAV3}$ denote ABLHs, peaks in each WAV profile; (d) K-means clustering analysis method. Black circles, red triangles, and blue x marks represent the different clusters, and the boundaries of the clusters ($h_{CLST1}$, $h_{CLST2}$) denote ABLH.**






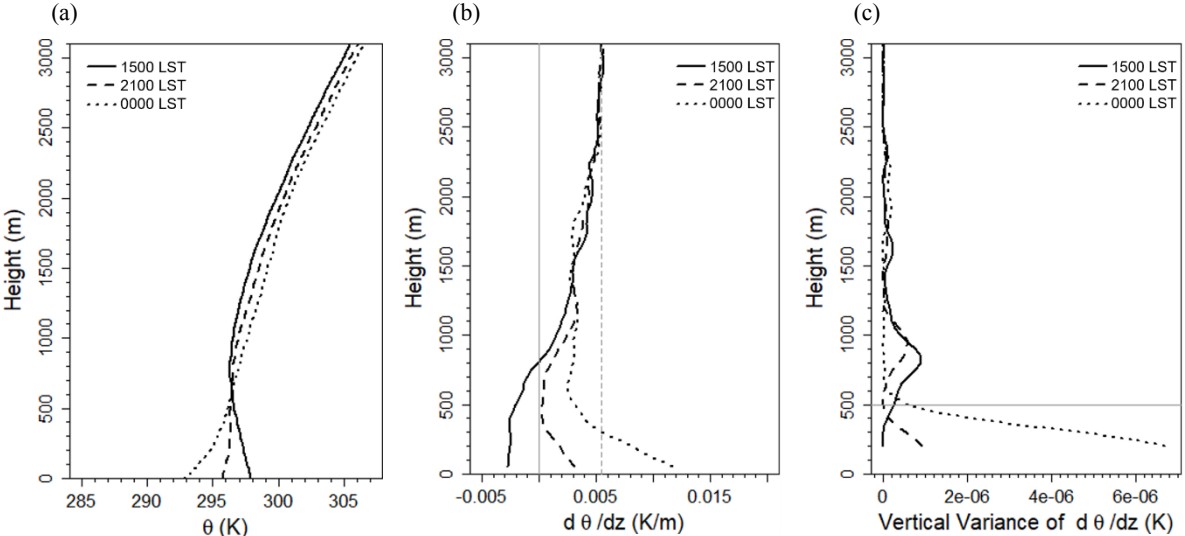

**Figure 4: Vertical profiles of (a) potential temperature, (b) gradient of potential temperature, (c) vertical variances of $d\theta/dz$ at 1500 LST (solid line), 2100 LST (dashed line) on 23, and 0000 LST (dotted line) on 24 September 2016. The vertical lines in (b) denote the theoretical lapse rate (solid gray line) and environmental lapse rate (dashed gray line). The altitude at which the vertical variance at 0000 LST on 24 September decreases sharply is denoted by gray line.**








**Figure 5: Flowchart of algorithm for ABLHs estimation from the vertical profile of backscattering coefficient obtained by a ceilometer.**





**Figure 6: Post-processing steps for determining the ABLH by ISABLE (ISABLE_ABLH): (a) time series of ABLH (CM_ABLH) without quality control; (b) applying $h_{SNR}$ threshold height and eliminating unreasonable value near the lens of ceilometer; (c) removing isolated ABLH using temporal discontinuity and DBSCAN; (d) The SBLHs estimated by microwave radiometer (MWR_SBLH) were merged, then post-processing is applied; (e) final ABLHs were determined the lowest layer.**





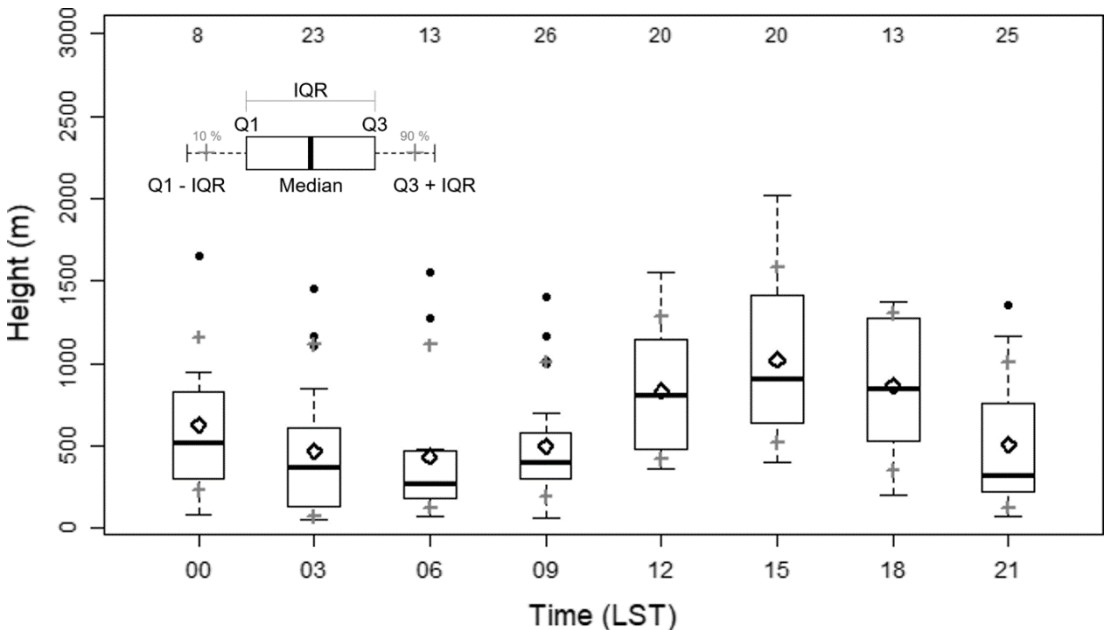

**Figure 7: Box plot of 3-h interval ABLHs estimated using the 148 radiosonde observed at Jungnang station for the year rom 2015 to 2018. The rhombus is the mean ABLH, the dot is outlier, the gray crosses represent 10% and 90% percentiles, respectively. IQR implies an interquartile range. The numbers at the top indicate the data frequency.**

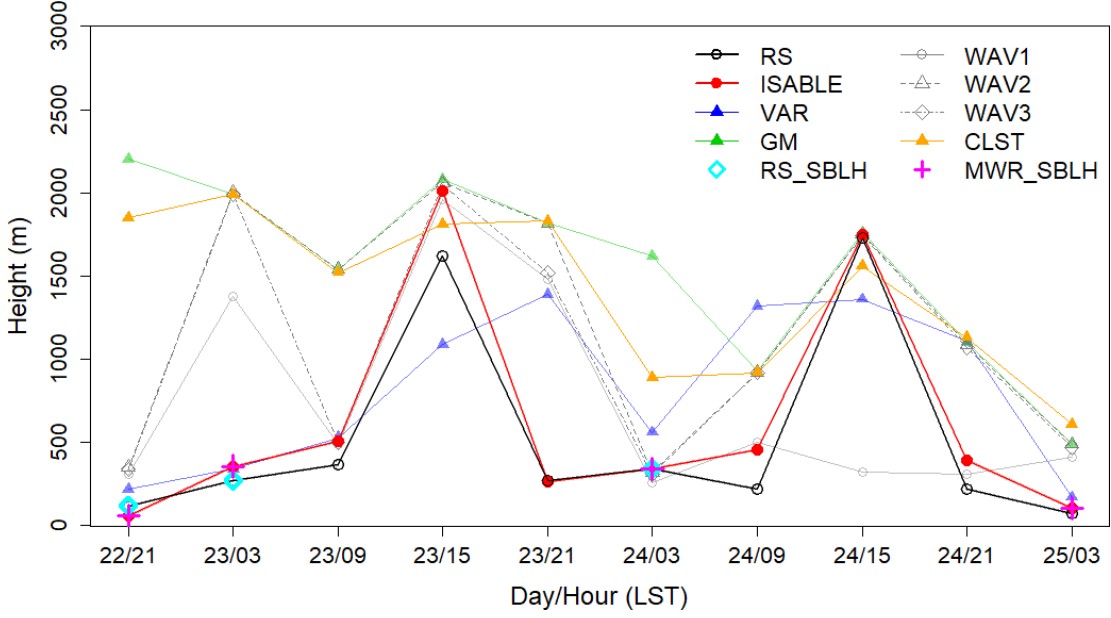

**Figure 8: Time series of ABLH estimated by radiosonde (RS), ISABLE, VAR, GM, WAV1, WAV2, WAV3, and CLST methodologies from 2100 LST on 22 to 0300 LST 25 September 2016. The SBLHs, estimated by radiosonde (RS_SBLH) and microwave radiometer (MWR_SBLH), are indicated at nighttime.**





(a)

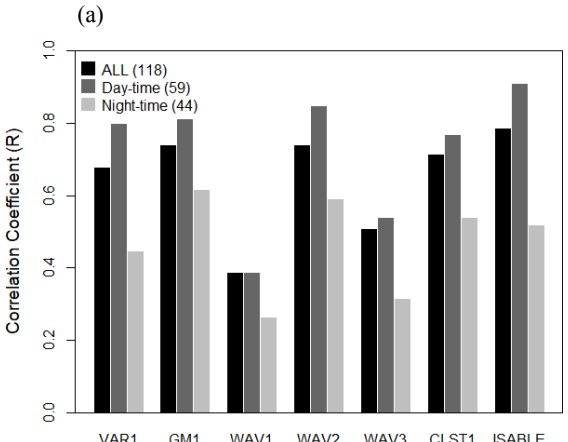

(b)

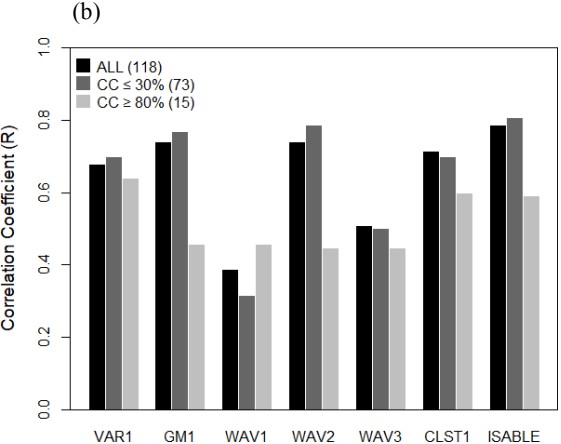

**Figure 9: Correlation coefficients between ABLH obtained from RS and those estimated by VAR1, GM1, WAV1, WAV2, CLST1, and ISABLE segregated by (a) daytime and nighttime, and (b) clear (cloud cover≤ 30%) and cloudy (cloud cover≥80%).**




(a)

(b)

**Figure 10: Box plots of hourly ABLHs estimated by ISABLE on (a) clear (cloud cover≤ 30%) and (b) cloudy (cloud cover ≥ 80%) cases for the period from November 2015 to October 2018.**



(a)

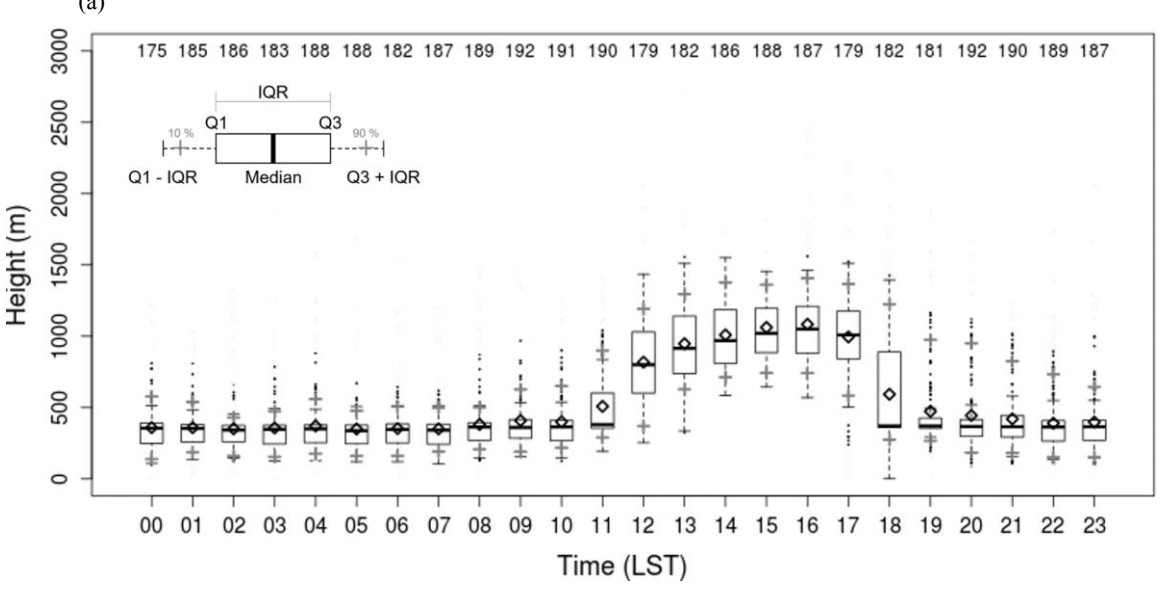

(b)

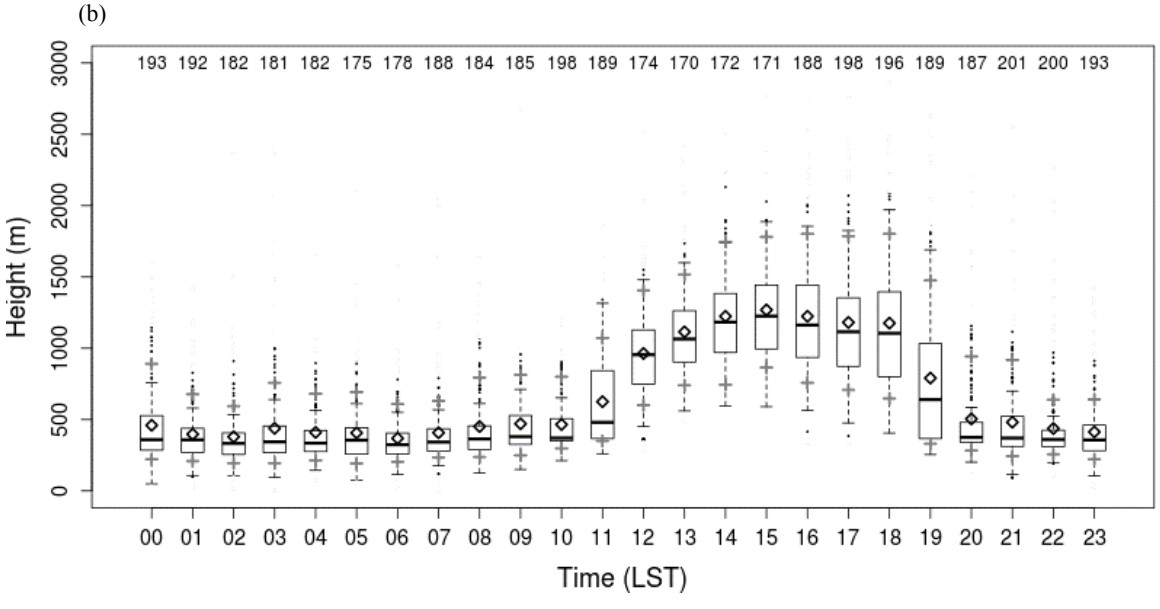


(c)

(d)

**Figure 11: Box plots of hourly ABLHs in (a) winter (December, January, February), (b) spring (March, April, May), (c) summer (June, July, August), and (d) autumn (September, October, November) for clear skies for the period from November 2015 to October 2018.**






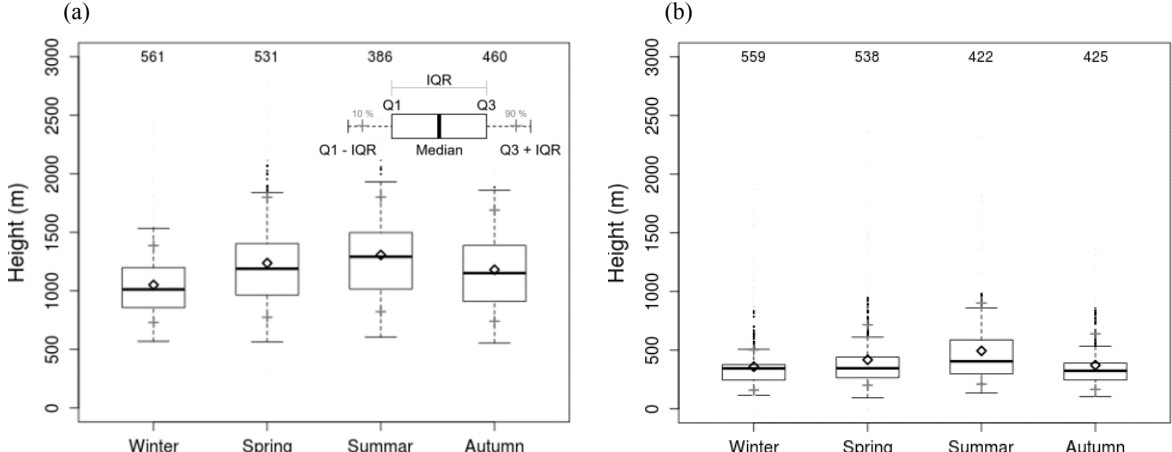

**Figure 12: Box plot of seasonal ABLH during the (a) daytime (1400 to 1600 LST) and (b) nighttime (0300 to 0500 LST) for the period from November 2015 to October 2018.**