# Peer review of "Integrated System for Atmospheric Boundary Layer Height Estimation (ISABLE) using a Ceilometer and Microwave Radiometer"

_Atmospheric Measurement Techniques, 2020_

## Referee Comment (RC1) · Anonymous Referee #1 · 27 May 2020

*Integrated System for Atmospheric Boundary Layer Height Estimation (ISABLE) using a Ceilometer and Microwave Radiometer. By Min et al.*

**General comments:**

This manuscript presents a method (ISABLE) for determining the atmospheric boundary layer (ABL) height using ceilometer profiles with stated improvement compared to previously well-established methodologies. The approach uses four existing methods (time-variance VAR, gradient GM, wavelet covariance WAV, and cluster analysis CLST methods) to identify multiple boundary layer candidates. It then uses a second cluster analysis approach to select the final ABL height. Microwave radiometer retrievals are then used to support stable nocturnal boundary layers. The ISABLE approach is then used in a long-term data set.

The manuscript's current structure and organization are adequate with some room for improvement. However, the text is often unclear with grammar oversights and language faults making it at times, difficult to follow. The most unique aspect of this methodology is the application of a cluster analysis that will select the ABL from initial results using multiple methodologies. Results show slight improvement in the ABLH using ISABLE when compared to the initial methodologies used. The manuscript does not provide discussions into the possible benefits from using the ISABLE approach, nor does it improve or expand on extensively studied challenges in ABLH detection using ceilometers.

The ISABLE methodology presented in this study should be further discussed and contrasted to previously published results as this is a field of extensive analysis. The results using the individual methods (VAR, GM, WAV, CLST) should be contrasted against published results using the same methodologies as the validation results of the individual methodologies in this manuscript are significantly lower than those in previously published studies (e.g. Cohn and Angevine, 2000; Morille et al., 2007; Münkel et al., 2007; Uzan et al., 2016; Caicedo et al., 2017). This invites questions about the application of the methodologies in this study.

Further, as ISABLE merges existing methodologies, it should be contrasted to similar algorithms which also merge various methodologies such as Lammert and Bösenberg (2006), Martucci et al. (2007), Morille et al. (2007), Di Giuseppe et al. (2012), Pal et al. (2013), Hicks et al. (2015), Geiß (2016), Peng et al. (2017), and Poltera et al. (2017). Similarly, although ISABLE in this case, slightly outperformed the individual methodologies in this study, it does not show improvements in the known challenges across all ABL ceilometer retrievals (ABL layer attribution, precipitation, lofted aerosol layers, low aerosol conditions, clouds, etc.). These challenges have been addressed in more recent methodologies which improve retrievals through various tracking tools (e.g. Lewis et al., 2013; Geiß, 2016; Poltera et al., 2017). Properly addressing the literature, would give further insight into the benefits or challenges of using ISABLE for ABLHs and may justify the suggested use of ISABLE.

The manuscript states improved results using ISABLE in comparison to VAR, GM, WAV, CLST methodologies. However, improvements in correlations when compared to sondes are slight and RMSE (above 300m) are substantial. These results and bias should be further discussed as a RMSE of 300m+ can be quite significant in the ABL.

For these reasons, the current state of the manuscript does not present substantial conclusions or new results that further our efforts in ceilometer ABL detection and should be considered for publication in AMT after major revisions.

**Specific comments:**

Section 2. Please provide references for ceilometer and MWR instrumentation. Additionally, the net radiometer is overlooked and left without description

Section 3.3. Please provide references to the entire methodology in this section.

Section 4.2.4: Was the cluster analysis applied to averaged $\bar{\beta}(z)$ or $\sigma_{\beta\,\mathrm{SNR}}$ profiles or else?

Section 4.3: MWR have also gone through extensive ABL retrieval evaluations and these should be discussed. The selection of the methodology should also be supported with further literature discussion. Additionally, is there a reason why MWR are only used for SBLH retrievals and not ML and RL heights?

Table 2 and 3: The difference between these two tables is not explained. Table 3 hints at a removal of "major error factors", how is this defined?

L23-24: Definitions of the ABL are unclear in this sentence. This reads as both the stable and residual layers are nighttime lofted layers. Please clarify.

L24: "By convection and turbulence" implies separate mechanisms independent of each other. Please review

L26: "Besides" is incorrectly used here.

L26-27: the reduced dilution volume of the SBL should also be noted as this can increase surface pollutant concentrations

L28-30: Although the ABL can influence pollutant concentration, it cannot be used to determine air pollutant concentrations as stated. The interactions between pollutant concentration and the ABL height are not the only drivers in air pollutant concentrations. Emission sources, transport, photochemistry etc. may have large impact in pollutant concentrations and must not be neglected.

L45: "back into the atmospheric aerosol" implies that the laser originated in atmospheric aerosol and returns back to its origin. Please revise

Equation (1): the variable N is not defined in the text

L126-130: Can you comment on how much of the improvement of $h_{\mathrm{SNR}}$ may be due to the applied averaging?

L162-164: It is unclear if residual layer heights are not being identified in radiosondes as the current text seems to imply aerosol profiles are sufficient for residual layer identification. Is the 'final ABLH' alternating between SBLs and RLs? Please clarify

L165-167: It is unclear what method was used in the final ABLH attributed to the SBL.

L177: Please describe how $\sigma_{\beta\,\mathrm{SNR}}$ profiles were smoothed

L193-195: How is this consistent with Emeis et al. (2008), please specify.

L207-208: How were these settings chosen?

L223-224: Please define what 'distribution' is divided

L284: Is this an artificial gap or a threshold marking?

L296-297: A follow up to the previous comment, does the final ABLH average include the added 150m or was this removed at some point? Please comment on how averaging the final cluster group can affect the ABLH, uncertainty, and validation.

L313-316: How was the 500m limit defined and what artifacts were observed? Please expand.

L333: This reads as the surface will only cool under clean air. Please review.

L335: the 'nocturnal heat island' effect is mentioned but not clearly connected to the results. Please expand on this effect and how at times (yet not always) it is responsible for high altitude outliers during nighttime.

L340-342: why was this time period chosen?

L369-390: No need to list all results already clearly presented in tables 2&3. Instead, insight into the possible culprits or conditions leading to the major findings should be presented and discussed.

L369-375: The clear and cloudy sky results (Figure 9b) shows very similar results across all methodologies (excluding WAV1&3) with a very small improvement in the ISABLE results. Can the benefits of ISABLE be expanded upon?

L388-390: This implies ABLHs were higher in altitude than other methodologies, yet Figure 9 refers to R, please review.

L392: It is unclear what 'merely 3(5) times higher' means. If SBL heights are a factor of 3-5 times higher than MWR, it is not a mere difference.

L 447: please specify these results correspond to daytime

L449: this is the first instance that R2 is used. Please revise for consistency and define all variables.

L452: This explains now the difference between Table 2 and 3. The removal of these "error" retrievals should be specified and described in the results section. Additionally, these should be clarified as manual removal of retrievals deemed inadequate and are not representative of the overall performance of ISABLE itself.

L458-460: The impact of ISABLE retrievals as "great potential in parameterizing vertical diffusion" and to "understand severe haze/smog events fumigated from the upper layer" is unsupported. Such statements require further discussion and supporting evidence.

**Technical:**
L72: repeated 'and' should be removed

L171: Should this be 'equation (7)'?

L439-440: "and compared to and verified". Should this be "are compared and verified against…"?

**References**:
Di Giuseppe, F., Riccio, A., Caporaso, L., Bonafé, G., Gobbi, G. P., and Angelini, F.: Automatic detection of atmospheric boundary layer height using ceilometer backscatter data assisted by a boundary layer model. Q. J. R. Meteorol. Soc.138:649–663. DOI:10.1002/qj.964, 2012

Geiß, A., Wiegner, M., Bonn, B., Schäfer, K., Forkel, R., von Schneidemesser, E., Münkel, C., Chan, K. L., and Nothard, R.: Mixing layer height as an indicator for urban air quality? Atmos. Meas. Tech. 10:2969–2988. DOI:10.5194/amt-10-2969-2017, 2017.

Hicks, M., Sakai, R., and Joseph, E.: The evaluation of a new method to detect mixing layer heights using Lidar observations. J. Atmos. Ocean. Technol., 32, 2041–2051, 2015.

Lewis, J. R., Welton, E. J., Molod, A. M., and Joseph, E.: Improved boundary layer depth retrievals from MPLNET, J. Geophys. Res. Atmos., 118, 9870-9879, doi:10.1002/jgrd.50570, 2013.

Martucci, G., Matthey, R., Mitev, V., and Richner, H. 2007. Comparison between Backscatter Lidar and Radiosonde Measurements of the Diurnal and Nocturnal Stratification in the Lower Troposphere, J. Atmos. Ocean. Technol. 24: 1231–1244, DOI:10.1175/JTECH2036.1

Morille, Y., Haeffelin, M., Drobinski, P., and Pelon, J.: STRAT: An Automated Algorithm to Retrieve the Vertical Structure of the Atmosphere from Single-Channel Lidar Data. J. Atmos. Ocean. Technol. 24:761–775. DOI:10.1175/JTECH2008.1, 2007.

Lammert, A. and Bösenberg, J.: Determination of the convective boundary-layer height with laser remote sensing. Boundary-Layer Meteorol. 119: 159–170. DOI:10.1007/s10546-005-9020-x, 2006.

Pal, S., Haeffelin, M., and Batchvarova, E.: Exploring a geophysical process based attribution technique for the determination of the atmospheric boundary layer depth using aerosol lidar and near-surface meteorological measurements. J. Geophys. Res. Atmos. 118: 9277–9295. DOI:10.1002/jgrd.50710, 2013.

Peng, J., Grimmond, C. S. B., Fu, X., Chang, Y., Zhang, G., Guo, J., Tang, C., Gao, J., Xu, X., and Tan, J.: Ceilometer based analysis of Shanghai's boundary layer height (under rain and fog free conditions). J. Atmos. Ocean. Technol. 34: 749- 764. DOI:10.1175/JTECH-D-16-0132.1, 2017.

Poltera, Y., Martucci, G., Collaud Coen, M., Hervo, M., Emmenegger, L., Henne, S., Brunner, D., and Haefele, A.: PathfinderTURB: an automatic boundary layer algorithm. Development, validation and application to study the impact on in situ measurements at the Jungfraujoch. Atmos. Chem. Phys. 17: 10051–10070. DOI:10.5194/acp-17-10051-2017, 2017.

Uzan, L., Egert, S., and Alpert, P.: Ceilometer evaluation of the eastern Mediterranean summer boundary layer height – first study of two Israeli sites, Atmos. Meas. Tech., 9, 4387–4398, doi:10.5194/amt-9-4387-2016, 2016.

---

## Referee Comment (RC2) · Anonymous Referee #3 · 3 Jul 2020

General Comment: The atmospheric boundary layer height (ABLH) is an important parameter to characterize the ABL and an important physical parameter in numerical simulations of the atmosphere and environmental assessments. It reflects turbulent mixing, convection, and other physical processes in the boundary layer, and affects the vertical distribution of substances and energy, such as heat, water vapor, and aerosols. In this manuscript, an integrated system was developed for ABLH estimation (ISABLE) using statistical techniques to produce one ABLH optimized by combining the determined ABLHs by previous methodologies. The results contribute to the reasonable determination of atmospheric boundary layer height. I think the work performed by the authors is useful and I recommend publication, after some revisions prior.

Specific points: 1, Line 26-27, 'Besides, when SBL exists at night, the lower atmosphere is stabilized and stagnant, and atmospheric diffusion does not occur in the lower layer', why? or weak diffusion? 2, Line 28-29, 'the ABL can be used as a meteorological factor to determine the air pollutant concentration', ABL should be ABLH. 3, Line 31-32, 'Many previous studies have developed various methodologies for determining ABLH, such as the ML height (MLH) and SBL height (SBLH).' It is better to change 'such as' into 'including'. 4, Line 35-36, it is better to using 'thermal turbulence and mechanical turbulence'. 5, Line 82-84, Section2, 'atmospheric attenuation and brightness temperature...' should be ' electromagnetic wave attenuation'. 6, Section3, Figure2, the noise is really less after pretreatment during the daytime, but the noise is more during the nighttime, why? 7, Section 4.1, Line 162-164, 'When determining the nocturnal SBLH, it is possible to estimate the SBLH using the vertical profile of the thermal parameter only because the turbulence or aerosol layer characteristics can be used to detect the residual-layer at night', please confirm the logical relationship. 8, Section 5.1, Line 334-335, 'ABLH of more than 1 km altitude appeared as outliers at nighttime', if possible, please show the data of the 'heat island phenomenon. 9, Section 5.3, Line 398-399, it is almost no difference for ABLH at clear skies (1202m) and cloudy skies (1085m), but the cloud cover is more difference, why? 10, Section 5., Line 403-404, The maximum seasonal mean ABLH was 1,268m in the Spring season (March, April, May), Please try to explain that using the net radiation data.

---

## Author Comment (AC1) · 29 Aug 2020

Response to referee #1 comments

Authors gratefully thank to referee #1 for his/her thorough reviews and valuable comments which would contribute to improve the manuscript. Authors have revised the manuscript to respond the referee's comments. Authors tried to improve the manuscript by clarifying the ambiguous expressions, and adding 3 Figures (Figs. 5, 10, and 11) with scatter plots between ISABLE_ABLH and conventional ABLHs with respect to time zones and cloud covers. The revised manuscript was edited by a professional Editing company. Major changes are marked in RED in the revised manuscript. I hope that

this manuscript will be accepted for the publication in AMT.

General comments:

On grammar:

- The revised manuscript was again edited by a professional editing company to improve grammatical and language faults.

On improvement and possible benefit of this study

- Poor performance was due to multiple factors, such as strong backscattering signals in the residual layer, presence of clouds, and too week backscattering signals. Overall, the performance of this study was found to be better than that of the conventional methods. It is difficult to estimate the consistent ABLHs under aforementioned atmospheric conditions. As the ABLH was estimated using as much data as possible, regardless of time or atmospheric conditions, their performances seemed to be somewhat lower (L499-501 and L505-507).

- The study station is located in an urban residential area with complex geography and topography, where can be affected by several types of local circulation such as sea-land breeze, mountain-valley breeze, and urban-rural breeze (L86-91). Moreover, the formation of evolution of SBL were not active over the station due to the heat release at nighttime by the heated materials during the daytime (L367-371).

- Nonetheless, it is found that the performance of ISABLE_ABLH was found to be better than that of the conventional methods (Section 5 and L499-501).

Further, as ISABLE merges existing methodologies, it should be contrasted to similar algorithms which also merge various methodologies.

- There were several previous studies on the merging algorithm such as Pal et al. (2013) and Hicks et al. (2015). Par et al. (2013) combined the gradient methods based on a first derivative of the Gaussian wavelet covariance analysis and the spatial/temporal variance method. Hicks et al. (2015) combined the error function-ideal profile method and wavelet covariance transform method to estimate ABLH (L67-69).

Similarly, although ISABLE in this case, slightly outperformed the individual methodologies in this study, it does not show improvements in the known challenges across all ABL ceilometer retrievals (ABL layer attribution, precipitation, lofted aerosol layers, low aerosol conditions, clouds, etc.).

- The possibilities of further improvement were discussed in L513-520 such that: Although ISABLE-estimated ABLH exhibited better performance than those estimated by the earlier conventional methodologies, there are still many limitations. In particular, ABLHs estimated from the ceilometer in the lower layer are not reliable due to near-range artifacts, especially under intense solar radiation. ABLHs at higher levels at nighttime could be supplemented by the temperature profile obtained by the MWR. ABLHs are challenging in terms of estimating under cloudy sky or precipitation, severe fog, and smog events. Since the ISABLE is in the early stage of development, it did not address the all known issues yet, such as precipitation, lofted aerosol layer, and too clean (little aerosol) condition. These limitations and drawbacks should be overcome by combining enough observation data, instrumental advances, and the corresponding improvements of ISABLE.

These challenges have been addressed in more recent methodologies which improve retrievals through various tracking tools. Properly addressing the literature, would give further insight into the benefits or challenges of using ISABLE for ABLHs and may justify the suggested use of ISABLE.

- Authors had reviewed the applicability of the recent methodologies within the range of the available data in Section 1 (Introduction).

- Several integrated methodologies are added in L67-70: Previous studies integrated multiple methodologies, i.e., Par et al. (2013) combined the gradient method based on a first derivative of the Gaussian wavelet covariance analysis and the spatial/temporal

variance method; and Hicks et al. (2015) combined the error function-ideal profile method and wavelet covariance method to estimate ABLH.

- Challenges are addressed in L71-76: Even though several methods have been developed, no consensus on a specific algorithm has been reached. Different methodologies provide different ABLHs with respect to weather conditions and phenomena. Under complicated ABL structures, the ABLH could be determined as different values according to the methodology used. Therefore, this study aims to develop an integrated system for ABLH estimation (ISABLE) to determine a single optimized ABLH with statistically significant results from several ABLH candidates.

The manuscript states improved results using ISABLE in comparison to VAR, GM, WAV, CLST methodologies. However, improvements in correlations when compared to sondes are slight and RMSE (above 300m) are substantial. These results and bias should be further discussed as a RMSE of 300m+ can be quite significant in the ABL.

- The MB and RMSE for nocturnal SBLH were as good as 6.7 m and 72 m, respectively, although the number of available data was not sufficient (L411-412).

- The performance of each methodology and ISABLE were discussed in L180-181, L369-371, L394-395, and L513-520: Firstly, radiosonde sounding data has a possibility to determine the SBLH as a residual layer due to the large variations of temperature and wind in the residual layer (L180-181); Secondly, the SBL over urban areas is not always developed because the sensible heat flux does not show strong negative values even at a clear night due to heat release by the heated materials during the daytime. So, formation and evolution of SBL were not active over compact urban surfaces such as the station (L369-371). Thirdly, ABLH via ceilometer is inclined to be estimated as a residual layer at nighttime due to the overlying aerosol layers. Scatter diagram shows that RLs during nighttime or cloud layers in daytime existed (Figs. 10-11) (L394-395). Finally, larger differences between ABLH_ISABLE and ABLH_RS were mainly due to the existence of residual layer and clouds as well as too clean (little aerosol)

atmosphere. Exact determination of ABLH above cases remains to be still challenges. These could be overcome by combining enough observation data, instrumental advances, and the corresponding improvement of ISABLE (L517-520).

Section 2: Please provide references for ceilometer and MWR instrumentation. Additionally, the net radiometer is overlooked and left without description.

- The references on ceilometer (Vaisala, 2010), MWR (RPG, 2015), and net radiometer (Kipp&Zonen, 2014) were added in L94, L98, and L106 in Section 2, respectively.

Section 3.3: Please provide references to the entire methodology in this section.

- Relevant references such as Cimini et al. (2006) and Holton and Hakim (2012) were addressed in L152 and L157 in Section 3.3.

Section 4.2.4: Was the cluster analysis applied to averaged (z) or $\sigma\beta$SNR profiles or else?

- The cluster analysis was applied to the backscattering coefficient $\beta$z in Figure 3d (L243-244).

Section 4.3: MWR have also gone through extensive ABL retrieval evaluations and these should be discussed. The selection of the methodology should also be supported with further literature discussion. Additionally, is there a reason why MWR are only used for SBLH retrievals and not ML and RL heights?

- The surface-based temperature inversion (SBI) using the radiosonde data was introduced in L179-185: SBLH is determined as a surface based temperature inversion (SBI) height at which the temperature decreases with height ($\Delta$T/$\Delta$z <0).

- In order to estimate SBLH using the MWR data, critical lapse rate (CLR) method was applied instead of SBI. To find the height affected by surface cooling, the critical lapse rate (CLR) method was proposed (L254-260). The reason why SBLHs obtained by the CLR method was explained and was compared with those obtained by the SBI method

(L261-278) in Section 4.3.

- But, during the RS intensive observation period, only 4 SBL were detected using the SBI methods from radiosonde (L292-293).

Minor comments:

L23-24: Definitions of the ABL are unclear in this sentence. This reads as both the stable and residual layers are nighttime lofted layers. Please clarify.

- In this study, the ABL is confined as a single layer, which is consisted of a mixed layer (ML) or a stable boundary layer (SBL) to exclude its complexity. Accordingly, the ABLH includes only a MLH or a SBL. The above explanation is added in L32 and L35.

L24: "By convection and turbulence" implies separate mechanisms independent of each other. Please review

- The sentence is rewritten as in L26-29: The ABL is repeated in a daily cycle with a mixed layer (ML) in daytime and a stable boundary layer (SBL) at nighttime. The former mixes air vertically via convection which results from surface heating or mechanical turbulence due to vertical wind shear, while the latter appears in the lower ABL, and a residual layer (RL) remains in the upper ABL without any external force.

L26: "Besides" is incorrectly used here.

- The term 'Besides' is removed in the revised manuscript.

L26-27: the reduced dilution volume of the SBL should also be noted as this can increase surface pollutant concentrations.

- The following explanation is added in L30-31: In the presence of well-developed SBL at night, air pollutants near the surface tend to be trapped inside the SBL because of the low vertical diffusivity, and their concentrations could increase sharply.

L28-30: Although the ABL can influence pollutant concentration, it cannot be used

to determine air pollutant concentrations as stated. The interactions between pollutant concentration and the ABL height are not the only drivers in air pollutant concentrations. Emission sources, transport, photochemistry etc. may have large impact in pollutant concentrations and must not be neglected.

- As you mentioned, ABL height is not only one drivers to determine the air pollutant concentration. But, the ABLH is an important factors to explain the vertical diffusion of air pollutants in the ABL. In the context, the sentence is rewritten as in L29-32: The ML is one of the essential meteorological factors that affects the vertical mixing of air pollutants. In the presence of well-developed SBL at night, air pollutants near the surface tend to be trapped inside the SBL because of the low vertical diffusivity, and their concentrations could increase sharply.

L45: "back into the atmospheric aerosol" implies that the laser originated in atmospheric aerosol and returns back to its origin. Please revise.

- The expression is clarified in L49-52: An aerosol lidar and a lidar-type ceilometer measure the intensity of signals which have been backscattered by atmospheric materials, such as aerosol, cloud, mineral dust. The intensity of backscattered signal at each level can be converted to backscattering coefficient at the level with several assumptions.

Equation (1): the variable N is not defined in the text.

- The variable N is defined in L131: N denotes the number of levels between 12 km and 15 km (N = 300).

L126-130: Can you comment on how much of the improvement of hSNR may be due to the applied averaging?

- Improvement of hSNR and the related explanation are added in L136-145: Strong noises with random backscattering coefficients were found at heights above 2,500 m throughout the day. When the shortwave radiation was intense during the daytime, the

noise was mainly due to sunlit scattering and low SNR values. Especially, in the presence of daytime clouds (1400 to 1600 LST), the SNR became smaller, and the hSNR became low. After pre-processing, noise signals at higher altitude have decreased with maintaining their main features in Fig. 2a (Fig. 2b). But vertical broadening at heights with intense signals was shown as a result of the moving average. And the mean hSNR became 331 m higher than the before. The pre-processing made the value much more stable, although under poor circumstances with strong radiation and daytime clouds. Also artifacts at high altitudes were mitigated.

L162-167: It is unclear if residual layer heights are not being identified in radiosondes as the current text seems to imply aerosol profiles are sufficient for residual layer identification. Is the 'final ABLH' alternating between SBLs and RLs? Please clarify. It is unclear what method was used in the final ABLH attributed to the SBL.

- Actually, it is not easy to detect a residual layer using the radiosonde sounding. This is because the vertical variations of temperature and wind in the RL can be more substantial compared to those in the SBL. Thus, the SBLH has been generally estimated using the methodologies with temperature inversion. In this study, the ABLHs were estimated with RiB in both daytime and nighttime, and if a SBL was formed at nighttime, the SBLHs were determined via the SBI method. Nonetheless, top of RL can be determined as a SBLH due to the effect of temperature and turbulence (Collaud Coen et al., 2014). The above explanation is added in L180-185.

L177: Please describe how $\sigma\beta$SNR profiles were smoothed

- The $\sigma_{(\beta\_VAR})$ profile was smoothed using a local quadratic polynomial regression to eliminate spurious variance peaks at small-scale fluctuations and above hSNR (Cleveland and Loader, 1996). The sentence is modified in L195-196.

L193-195: How is this consistent with Emeis et al. (2008), please specify.

- The sentence is clarified in L211-212: The fact that hGM is slightly higher than hIPM,

and lower than hLGM is consistent with the findings of previous studies (e.g., Emeis et al., 2008).

L207-208: How were these settings chosen?

- The settings were just followed by those of de Haij et al. (2006; 2007). The reference is added in L227.

L223-224: Please define what 'distribution' is divided

- The term 'distribution' was changed to 'height' in L242.

L284, 296-297: Is this an artificial gap or a threshold marking? A follow up to the previous comment, does the final ABLH average include the added 150m or was this removed at some point? Please comment on how averaging the final cluster group can affect the ABLH, uncertainty, and validation.

- The reasons for choosing the 150 m and more explanation are added in L310-314: The minimum distance between the nearest two ABLH candidates was set to 150 m. The reason is that the typical thickness of a well-defined entrainment zone was reported to be between 100 and 300 m (e.g., Angevine et al., 1994). If there were multiple peaks chosen using an individual methodology within 150 m interval, the remaining peaks except for the most significant one were removed from the ABLH candidates for the method.

L313-316: How was the 500m limit defined and what artifacts were observed? Please expand.

- The following sentence is added in L139-140 and L346-351: As shown in Fig. 2a, it was found that backscattering signals were weakened at about 120 m high and 400-500 m during the daytime with intense solar radiation. Due to the weakened signal by instrumental reason, the 400-500 m was often estimated as an ABLH. So, ABLHs below 500 m at the time were assumed to be unreasonable and were neglected.

L333: This reads as the surface will only cool under clean air. Please review.

- More explanation is added in L367-371: The SBL over rural areas is well developed over rural surfaces via surface cooling through earth radiation at night, especially under clear skies. However, that over urban areas is not always developed because the sensible heat flux in urban areas does not always show strong negative values even at a clear night due to heat release by the heated materials during the daytime (Hong et al., 2013; Park et al., 2014). So formation and evolution of SBL were not active over compact urban surfaces such as Jungnang station.

L335: the 'nocturnal heat island' effect is mentioned but not clearly connected to the results. Please expand on this effect and how at times (yet not always) it is responsible for high altitude outliers during nighttime.

- The outliers above 1 km results from a RL or cloud layer. When the nocturnal SBLH was estimated using aerosol-related variables at nighttime, the ABLH can be often detected as a top of residual layer. This is because the vertical variations of the backscattering signal in the RL can be more substantial compared to those in the SBL. The above explanations are in L180-182 and L372-373.

L340-342: why was this time period chosen?

- The period is chosen considering the consecutive observation period and missing ratios among available radiosonde sounding data. The period corresponds to the longest observation period with an interval of 3 h and without any missing data among available radiosonde data. The above explanation is added in L378-379.

L369-390, L369-375: No need to list all results already clearly presented in tables 2&3. Instead, insight into the possible culprits or conditions leading to the major findings should be presented and discussed. The clear and cloudy sky results (Figure 9b) shows very similar results across all methodologies (excluding WAV1&3) with a very small improvement in the ISABLE results. Can the benefits of ISABLE be expanded

upon?

- The paragraph is fully removed and rewritten according to the suggestion. In order to find insight, analyses on ABLH with respect to 4 time zones (sunrise, daytime, sunset, nighttime) and cloud covers (clear and cloudy) are added in Section 5.2. As a result, the improvement of ISABLE was evident during transition time, sunrise and sunset, while the verification score was not good at nighttime. It seems to be the difference of SBLH estimation between the radiosonde and microwave radiometer, and further analysis is required on this problem. In this study, it is considered difficult to analyze due to the limitation of the number of radiosonde observation at nighttime. The above explanation is added in L378-412, Table 2-3, Figures 10 and 11.

- The variance (VAR) and the cluster (CLST) methods showed the best performances during the daytime. The scatter distribution of GM, WAV2, and CLST at sunrise, sunset, and nighttime could be fitted to two different linear functions. In cases where symbols were plotted below the trend line, RLs during nighttime or cloud layers in daytime existed at the layer. ISABLE showed significant improvement near sunrise and sunset time, but showed a lower correlation with the individual methods in nighttime because ABLH was often underestimated, as compared with RS. There were only four SBLH estimations via RS, while 24 SBLH were observed via MWR, which resulted in significantly lower ISABLE performance at nighttime, as compared with those of the four methodologies. The performances of WAV1 and 3 were significantly poorer than those of other individual methodologies. The shorter dilation used in WAVE1 seems to be unsuitable for estimating the ABLH, and it might affect the ABLH of WAV3 (L392-403).

Table 2 and 3, L452: The difference between these two tables is not explained. Table 3 hints at a removal of "major error factors", how is this defined? This explains now the difference between Table 2 and 3. The removal of these "error" retrievals should be specified and described in the results section. Additionally, these should be clarified as manual removal of retrievals deemed inadequate and are not representative of the overall performance of ISABLE itself.

- Tables 2 is rewritten from a point of view of four time zone (sunrise, daytime, sunset, nighttime) using the data listed in Table 1. On a while, Table 3 is based on the last three data listed in Table 1. It is because cloud cover was missing in the first data (23 to30 November 2015) in Table 1. Minor statistical errors were corrected in revised Table 3. The above explanation is added in L403-405.

- The explanation in Section 6 is moved to in L403-405 in Section 5.2.

- The limitation of this study is added in L513-520: Although ISABLE-estimated ABLH exhibited better performance than those estimated by the earlier conventional methodologies, there are still limitations. In particular, ABLHs estimated from the ceilometer in the lower layer are not reliable due to near-range artifacts, especially under intense solar radiation. ABLHs at higher levels at nighttime could be supplemented by the temperature profile obtained by the MWR. ABLHs are challenging in terms of estimating under cloudy sky or precipitation, severe fog, and smog events. Since the ISABLE is in the early stage of development, it did not address the known issues yet, such as precipitation, lofted aerosol layers, and too clean (little aerosol) conditions. These limitations and drawbacks should be overcome by combining enough observation data, instrumental advances, and the corresponding improvement of ISABLE.

L458-460: The impact of ISABLE retrievals as "great potential in parameterizing vertical diffusion" and to "understand severe haze/smog events fumigated from the upper layer" is unsupported. Such statements require further discussion and supporting evidence.

- The expression is not supported by clear evidences yet, and is removed in the revised manuscript.

Please also note the supplement to this comment:
https://amt.copernicus.org/preprints/amt-2020-18/amt-2020-18-AC1-supplement.pdf

---

## Author Comment (AC2) · 29 Aug 2020

**Response to referee #2 comments**

Authors gratefully thank to referee #2 for his/her thorough reviews and valuable comments which would contribute to improve the manuscript. Authors have revised the manuscript to respond the referee's comments. Authors tried to substantially improve the manuscript by clarifying the ambiguous expressions, and adding 3 Figures (Figs. 5, 10, and 11) with scatter plots between ISABLE\_ABLH and conventional ABLHs with respect to time zones and cloud covers. The revised manuscript was edited by a professional Editing company. Major changes are marked in RED in the revised manuscript.

I hope that this manuscript will be accepted for the publication in AMT.

Specific points:

1, Line 26-27, 'Besides, when SBL exists at night, the lower atmosphere is stabilized and stagnant, and atmospheric diffusion does not occur in the lower layer', why? or weak diffusion?

-The sentence is rewritten as "In the presence of well-developed SBL at night, air pollutants near the surface tend to be trapped inside the SBL due to the low vertical diffusivity, and their concentrations could increase sharply (L30-32).

2, Line 28-29, 'the ABL can be used as a meteorological factor to determine the air pollutant concentration', ABL should be ABLH.

- The term 'ABL' is corrected to the term 'ABLH' (L33).

3, Line 31-32, 'Many previous studies have developed various methodologies for determining ABLH, such as the ML height (MLH) and SBL height (SBLH).' It is better to change 'such as' into 'including'.

-The term 'such as' is changed to the term 'including' (L35).

4, Line 35-36, it is better to using 'thermal turbulence and mechanical turbulence'.

- The sentence is changed to '~which includes the thermal turbulence term generated by surface heating as well as the mechanical turbulence term arising from the vertical wind shear.' (L39-40).

5, Line 82-84, Section2, 'atmospheric attenuation and brightness temperature...' should be' electromagnetic wave attenuation'.

- The expression is rewritten as 'atmospheric attenuation and brightness temperature from electromagnetic radiation' (L96-97).

6, Section3, Figure2, the noise is really less after pretreatment during the daytime, but
the noise is more during the nighttime, why?

- The strong noise with random backscattering coefficient was seen at heights above 2,500 m throughout the day in Fig. 2a. After pre-processing, noise signals at higher altitudes have decreased with maintaining their main features in Fig. 2a. But, vertical broadening at heights with strong signals was shown as a result of moving averaging. The above sentences were added in L136-145.

7, Section 4.1, Line 162-164, 'When determining the nocturnal SBLH, it is possible to estimate the SBLH using the vertical profile of the thermal parameter only because the turbulence or aerosol layer characteristics can be used to detect the residual-layer at night', please confirm the logical relationship.

- The above sentence is clarified as 'Actually, it is not easy to detect a residual layer using radiosonde sounding. This is because the vertical variations of the moisture and the wind in the residual layer can be more substantial compared to those in the SBL (L179-181)

8, Section 5.1, Line 334-335, 'ABLH of more than 1 km altitude appeared as outliers at nighttime', if possible, please show the data of the 'heat island phenomenon.

- Main factors for nocturnal urban heat islands are heat release by the heated materials during the daytime in urban areas, mentioned in the previous studies. The following expressions are added in L366-370: The SBL over rural areas is well developed over rural surfaces via surface cooling from earth radiation at night, especially under clear skies. However, that over urban area is not always developed because the sensible heat flux in urban areas does not always show strong negative values even at a clear night due to heat release by the heated materials during the daytime. So formation and evolution of SBL were not active over compact urban surfaces such as Jungnang station.

9, Section 5.3, Line 398-399, it is almost no difference for ABLH at clear skies (1202m)
and cloudy skies (1085m), but the cloud cover is more difference, why?

- The complementary explanation is added in L419-422: The period mean hourly maximum ABLH was 1,220 m at 1600 LST on clear skies, while it was 1,090 m at 1500 on cloudy skies. Diurnal pattern and mean of ABLH on clear skies seemed to be similar to those on cloudy skies. But, median of ABLH was 1,170 m at 1600 LST on clear skies, 210 m higher than that (960 m) at 1500 LST on cloudy skies. Variances of ABLH on cloudy skies were also larger than those on clear skies.

10, Section 5., Line 403-404, The maximum seasonal mean ABLH was 1,268m in the Spring season (March, April, May), Please try to explain that using the net radiation data.

- Some errors in cloud cover calculation were found in 2015 and 2016, so the ABLHs were re-evaluated. The diurnal variation of ABLH was compared with that of net radiation in Fig. 15. Theoretically, the surface is heated from the time when net radiation becomes positive, and an ML evolves to balance the energy provided from the surface during the positive net radiation with the energy consumed to heat the overlying air volume. In reality, the ABL started to evolve from 3 h after the positive net radiation. The peak of net radiation occurred at 1200 LST, while the peak of ABLH occurred at about 1600 LST. The ABLH declined rapidly at 1 to 2 h before the negative net radiation. The net radiation in MAM was similar to that in JJA, and larger than that in SON, while the ABLH in MAM was similar to that in JJA, and larger than that in SON. The above explanation is added in L450-455.

Please also note the supplement to this comment: https://amt.copernicus.org/preprints/amt-2020-18/amt-2020-18-AC2-supplement.pdf AMTD

---

## Referee Report (RR1)

The reviewer commends the authors for the hard work applied to this manuscript that has resulted in a much-improved work.

General Comments:
The authors were asked to compare with previous studies of individual methodologies. Although more references were added, the fact that previous studies of individual methodologies show better results those than shown in the manuscript was not addressed. The methodology presented here does not address the impact of clouds/precipitation and individual methodologies do not use post-processing application which is likely the reason why individual methodologies perform worse than those in literature. By consider the previous literature, the authors should note that ISABLE uses existing methodologies and then applies a cluster technique to addresses layer attribution in post processing. As all other individual methodologies are not applied with any improvements for layer attribution, consider stating that ISABLE *improved* PBLH retrievals due to the layer attribution techniques. Stating the ISABLE was superior to individual methodologies is misleading as individual methodologies were not aided by post processing techniques as ISABLE was.

Specific Comments:
L26: consider the term mixed layer. As a dynamic layer, the ML is not under a finite state.

L53: Correct to "In a ML"

L66: Studies have shown that the EKF technique is sensitive to low SNR and to indeed require further long-time averaging and range smoothing. Please revise this statement.

L180-185: It is stated that defining the RL using radiosonde profiles is difficult due to large variations in temperature and wind profiles. L184-185 later states that the top of the RL is determined as the SBLH due to large variations in temperature and turbulence. At first glance it seems contradicting, please expand on this. Were these occurrences during certain conditions, or errors from the retrieval method?

L367: Provide uncertainties here

L368-371: Please revise, repetition and grammatical errors make these statements unclear. What are rural surfaces? What does 'that' refer to? What is defined as a compact urban surface?

L371: As previously stated "However, that over urban areas is not always developed" in L368-369, L371 should state that 'SBL were not always active'.

L372: "which were determined using the residual layer or clouds" should this state "which were identified as the residual layer or clouds"?

L380: R symbol needs to be corrected

L382: Is this still referring to ISABLE results or all ABLH results? I suspect this is referring to GM, CLST, and WAV results. Please clarify. If this refers to ISABLE please note that the residual layer contains the remains of the previous day's mixing layer at similar heights. With peak daytime mixing layers in Figure 9 at ~1600 (RS), residual layers signals would be expected in similar heights. Figure 9 shows ISABLE overestimations not exceeding ~500m therefore, how can the overestimation of ISABLE during nighttime be attributed to

residual layer signals? Instead, it is likely due to additional stratification of the PBL and therefore additional lofted aerosol layer during nighttime.

L391: MB was not defined in text

L393: Please clarify what "The scatter distribution of GM, WAV2, and CLST at sunrise, sunset, and nighttime could be fitted to two different linear functions" means

L394: "In cases where symbols were plotted below the trend line (dashed line), RLs during nighttime or cloud layers in daytime existed at the layer". Please clarify the effect of "cloud layers [existing] at the layer"

L399: As stated in L367-371, the urban effect will impact the SBL detection from RS. Consider clarifying this statement by adding this overestimation effect on RS heights that lead to an underestimation when compared to ABLHs.

L401: Use WAV3

L405-408: Similar as above, please clarify the statement "could be fitted to two different linear lines"

L419-420: "The ABLHs for clear skies were significantly higher than those for cloudy skies during the daytime, however, the difference was not as significant during the **daytime**" Is this repetitive or a typo?

L479-481: What was further verified?

L483-484: See general comments above. The 'superior' performance of ISABLE is largely due to the addition of post-processing techniques. As individual methodologies are not applied with any post-processing techniques, it is not correct to state a superior performance. Instead, the large improvement that was seen with the ISABLE post-processing technique should be highlighted.

---

## Author Response (AR2)

Response to referee #1 comments

Authors gratefully thank to referee #1 for his/her thorough reviews and valuable comments which would contribute to improve the manuscript. Authors have revised the manuscript to respond the referee's comments. Major changes are marked in **red** in the revised manuscript. I hope that this manuscript will be accepted for the publication in AMT.

**General Comments:**

The authors were asked to compare with previous studies of individual methodologies. Although more references were added, the fact that previous studies of individual methodologies show better results those than shown in the manuscript was not addressed.

> Most conventional methodologies have been verified for daytime clear skies during the several days. While this study tried to attempt to consider all available weather conditions including cloudy sky during the 4 years. Therefore, the verification values were lower than those for the previous studies. The above expression is added in L515-517.

The methodology presented here does not address the impact of clouds/precipitation

and individual methodologies do not use post-processing application which is likely the reason why individual methodologies perform worse than those in literature. By consider the previous literature, the authors should note that ISABLE uses existing methodologies and then applies a cluster technique to addresses layer attribution in post processing. As all other individual methodologies are not applied with any improvements for layer attribution, consider stating that ISABLE improved PBLH retrievals due to the layer attribution techniques. Stating the ISABLE was superior to individual methodologies is misleading as individual methodologies were not aided by post processing techniques as ISABLE was.

> Not cloud but precipitation is removed before data processing in this studies. The effects on clouds were explained in **Fig. 11** and **Table 3**.

> The core post-processing technique of this study is to cluster for multiple layers. The ABLH calculated by the methodology used in the previous studies are not multi-layer but single layer. On the other hand, ISABLE can cluster by calculating several ABLHs using four methodologies, and improved ABLH calculation performance through statistical post-processing.

> A description of multiple ABLH estimation using four different methodologies and performance improvement through statistical post-processing is given in **L474-475**: The ISABLE developed in this study integrated the conventional ABLH estimation methodologies to produce optimal ABLH and applied statistical post-processing techniques to improve accuracy.

**Specific Comments:**

L26: consider the term mixed layer. As a dynamic layer, the ML is not under a finite state.

> The term 'mixed layer (ML)' is changed to the term 'a well-mixed layer (ML) or a convective boundary layer (CBL)'. (**L26**).

L53: Correct to "In a ML"

> The term 'In ML' is corrected to the term 'In a ML' in **L54.**

L66: Studies have shown that the EKF technique is sensitive to low SNR and to indeed require further longtime

averaging and range smoothing. Please revise this statement.

> The term 'except for low SNR' is added at the end of the statement in L68.

L180-185: It is stated that defining the RL using radiosonde profiles is difficult due to large variations in temperature and wind profiles. L184-185 later states that the top of the RL is determined as the SBLH due to large variations in temperature and turbulence. At first glance it seems contradicting, please expand on this. Were these occurrences during certain conditions, or errors from the retrieval method?

> The term 'RL' is changed to the term 'SBLH' in L181.

L367: Provide uncertainties here

> The uncertainty was analyzed in terms of IQR (interquartile range) and added in L368-371 such that: At night, the mean ABLHs were determined as around 500 m, and outliers appeared above 1 km, which were identified as the RL or clouds (Fig. 8). The interquartile range (IQR; Q3 − Q1) showed the minimum value (268 m) at 0900 LST and the maximum (740 m) at 1800 LST. Overall, ABLHs were concentrated in the lower layer at night, and the IQR values increased as the ML developed after sunrise.

L368-371: Please revise, repetition and grammatical errors make these statements unclear. What are rural surfaces? What does 'that' refer to? What is defined as a compact urban surface?

> The repetition is removed, and ambiguous expression or grammatical errors are corrected in L372-375: The SBL over rural areas such as grass or cropfield is well developed due to active radiative cooling at night, especially under clear skies. On a while, the radiative cooling over urban areas was not always active because of heat storage by urban materials and anthropogenic heat by energy use (Hong et al., 2013; Park et al., 2014). As a result, formation and evolution of SBL were not active over dense urban areas such as Jungnang station.

L371: As previously stated "However, that over urban areas is not always developed" in L368-369, L371 should state that 'SBL were not always active'.

> The sentence is changed to 'As a result, formation and evolution of SBL were not active over dense urban areas such as Jungnang station' in **L374-375.**

L372: "which were determined using the residual layer or clouds" should this state "which were identified as the residual layer or clouds"?

> The sentence is changed to 'which were identified as the RL or clouds' in **L369.**

L380: R symbol needs to be corrected

> It is corrected as 'R' in **L381.**

L382: Is this still referring to ISABLE results or all ABLH results? I suspect this is referring to GM, CLST, and WAV results. Please clarify. If this refers to ISABLE please note that the residual layer contains the remains of the previous day's mixing layer at similar heights.

> The statement is clarified in L382-385 such that: The ABLHs from ISABLE as well as ceilometer-based methods (GM, WAV2, WAV3, and CLST) were similar to those by RS during the daytime, however, the ABLHs from the former appeared at higher levels than those from the latter during the nighttime.

With peak daytime mixing layers in Figure 9 at ~1600 (RS), residual layers signals would be expected in similar heights. Figure 9 shows ISABLE overestimations not exceeding ~500m therefore, how can the overestimation of ISABLE during nighttime be attributed to residual layer signals? Instead, it is likely due to additional stratification of the PBL and therefore additional lofted aerosol layer during nighttime.

> As can be seen in Fig.7, it is difficult to judge this as a residual layer signal. The difference between RS and ISABLE at 1500 LST on 23 is due to cumulus cloud. The following paragraph is added in **L385-395**: ISABLE tried to complement the shortcomings by integrating the four methodologies through considering the SBL using a vertical temperature from MWR at night. The maximum ABLHs during daytime appeared at 1600 LST on 23, the RS and the ISABLE estimated ABLHs of 1,620 m, 2,009 m, respectively. At this time, a cumulus cloud was formed over the top of ABL due to strong convection, and the cloud base height observed by the ceilometer was 1,910 m. The ABLHs estimation results showed that RS was below the cloud, while ISABLE and individual methodologies (GM: 2,080 m, WAV2: 2,060 m, WAV3: 2,050 m) detected ABLHs as the cloud. In the presence of clouds, the $Ri_b$ method tends to detect the lower layer of the cloud, where the temperature profile changes rapidly. The GM and WAV2 methods using the ceilometer determine the ABLHs as the top of loud layer because of strong negative gradient of backscattering coefficient, whereas the CLST can detect both the bottom and top of cloud layer. In ISABLE, the effect of clouds is compensated for averaging multiple heights determined by individual methodologies. However, the ISABLE still has limitations in the presence of thick clouds.

L391: MB was not defined in text

> The MB means mean bias. The definition is added in **L382**.

L393: Please clarify what "The scatter distribution of GM, WAV2, and CLST at sunrise, sunset, and nighttime could be fitted to two different linear functions" means

>> The sentence is clarified in 405 such as: The scatter distribution could be divided into two groups with different linear functions.

L394: "In cases where symbols were plotted below the trend line (dashed line), RLs during nighttime or cloud layers in daytime existed at the layer". Please clarify the effect of "cloud layers [existing] at the layer"

> Effect of cloud is explained in **L390-395**: In the presence of clouds, the $Ri_b$ method tends to detect the lower layer of the cloud, where the temperature profile changes rapidly, as the ABLH. The GM and WAV2 methods using the ceilometer detect the upper layer of the cloud because the height at which the backscattering coefficient rapidly decreases is determined by ABLH, whereas the CLST can detect both the lower and upper layer of the cloud. In ISABLE, the effect of clouds is compensated for averaging multiple heights determined by individual methodologies. However, the ISABLE still has limitations in the presence of thick clouds.

L399: As stated in L367-371, the urban effect will impact the SBL detection from RS. Consider clarifying this statement by adding this overestimation effect on RS heights that lead to an underestimation when compared to ABLHs.

> It is an underestimation of the frequency of occurrence, not the height. Since there is a possibility of misunderstanding, it has been modified to be more precise in **L410-411**: Anthropogenic heat release from urban materials could be one reason for detecting less number of SBLHs at night.

L401: Use WAV3

> It is modified to "WAV3" in **L412.**

L405-408: Similar as above, please clarify the statement "could be fitted to two different linear lines"

> The sentence is clarified in L419 such as: The scatter plots could be classified into two groups with different linear lines.

L419-420: "The ABLHs for clear skies were significantly higher than those for cloudy skies during the daytime, however, the difference was not as significant during the **daytime**" Is this repetitive or a typo?

> It is corrected as "nighttime" in **L432.**

L479-481: What was further verified?

> It is corrected to ". Furthermore, the ISABLE was verified through the separation of the data into four time zones" in **L492-493.**

L483-484: See general comments above. The 'superior' performance of ISABLE is largely due to the addition of post-processing techniques. As individual methodologies are not applied with any post-processing techniques, it is not correct to state a superior performance. Instead, the large improvement that was seen with the ISABLE post-processing technique should be highlighted.

> The individual methodologies of previous studies determine one layer, and even if multiple layers are additionally determined, the cluster classification of post-processing is not possible. On the other hand, ISABLE can cluster by calculating several ABLHs using four methodologies, and improved ABLH calculation performance through statistical post-processing.

> A description of multiple ABLH estimation using four different methodologies and performance improvement through statistical post-processing is given in **L474-475**: The ISABLE developed in this study integrated the conventional ABLH estimation methodologies to produce optimal ABLH and applied statistical post-processing techniques to improve accuracy.